# Neural excitability and sensory input determine intensity perception with opposing directions in initial cortical responses

Tilman Stephani[1,2]*, Alice Hodapp[1†], Mina Jamshidi Idaji[1,2,3], Arno Villringer[1,4,5], Vadim V Nikulin[1,6]*

[1]Department of Neurology, Max Planck Institute for Human Cognitive and Brain Sciences, Leipzig, Germany; [2]International Max Planck Research School NeuroCom, Leipzig, Germany; [3]Machine Learning Group, Technical University of Berlin, Berlin, Germany; [4]Berlin School of Mind and Brain, Humboldt-Universität zu Berlin, Berlin, Germany; [5]Clinic for Cognitive Neurology, University Hospital Leipzig, Leipzig, Germany; [6]Institute for Cognitive Neuroscience, National Research University Higher School of Economics, Moscow, Russian Federation

*For correspondence:
stephani@cbs.mpg.de (TS);
nikulin@cbs.mpg.de (VVN)

Present address: †Now at Department of Psychology, University of Potsdam, Potsdam, Germany

Competing interest: The authors declare that no competing interests exist.

**Abstract** Perception of sensory information is determined by stimulus features (e.g., intensity) and instantaneous neural states (e.g., excitability). Commonly, it is assumed that both are reflected similarly in evoked brain potentials, that is, larger amplitudes are associated with a stronger percept of a stimulus. We tested this assumption in a somatosensory discrimination task in humans, simultaneously assessing (i) single-trial excitatory post-synaptic currents inferred from short-latency somatosensory evoked potentials (SEPs), (ii) pre-stimulus alpha oscillations (8–13 Hz), and (iii) peripheral nerve measures. Fluctuations of neural excitability shaped the perceived stimulus intensity already during the very first cortical response (at ~20 ms) yet demonstrating opposite neural signatures as compared to the effect of presented stimulus intensity. We reconcile this discrepancy via a common framework based on the modulation of electro-chemical membrane gradients linking neural states and responses, which calls for reconsidering conventional interpretations of brain potential magnitudes in stimulus intensity encoding.

## Introduction

Even for the very same stimulus, the brain's response differs from moment to moment. This has been explained by ever-changing neural states (*Arieli et al., 1996*), with behaviorally relevant consequences (*Waschke et al., 2021*). Specifically, these changes of neural states are often conceptualized as fluctuations of cortical excitability (*Jensen and Mazaheri, 2010*; *Klimesch et al., 2007*; *Romei et al., 2008*). In the human brain, a commonly hypothesized marker of cortical excitability is oscillatory activity in the alpha band (8–13 Hz), which can be measured with electro- and magnetoencephalography (EEG/MEG). This marker has been associated with modulations of a stimulus' percept in various sensory domains including the visual (*Busch et al., 2009*; *Iemi et al., 2017*), auditory (*Müller et al., 2013*), and somatosensory domain (*Baumgarten et al., 2016*; *Craddock et al., 2017*; *Forschack et al., 2020*). According to the baseline sensory excitability model (BSEM; *Samaha et al., 2020*), higher alpha activity preceding a stimulus indicates a generally lower excitability level of the neural system, resulting in smaller stimulus-evoked responses, which are in turn associated with a lower detection rate of near-threshold stimuli but no changes in the discriminability of sensory stimuli (since neural

noise and signal are assumed to be affected likewise). On a cellular level, such excitability modulations may be reflected in changes of membrane potentials (*Castro-Alamancos, 2009*), which may occur in an oscillatory manner (*Lakatos et al., 2005*) and shift the threshold for incoming sensory information to be processed further downstream in the neural system. This notion has been further supported by monkey studies showing that higher oscillatory activity within the alpha band is associated with a lower neural firing rate (*Bollimunta et al., 2011*; *Haegens et al., 2011*).

However, it remains unclear up to now whether the influence of instantaneous excitability on perceptual processes can be generalized to the intensity perception of stimuli per se (i.e., beyond the sensory threshold) – which would have far-reaching implications for a wide variety of studies in the field of perception. Moreover, if such modulation indeed occurs, the question remains: At which stage of the neural response cascade do instantaneous excitability changes begin to interact with the sensory input in order to shape the brain's response in a behaviorally relevant way?

A unique opportunity to non-invasively measure instantaneous excitability changes of neurons involved in the first cortical response to sensory stimuli in humans is offered by the N20 component of the somatosensory evoked potential (SEP) as measured with EEG: The N20 component, a negative deflection after around 20 ms at centro-parietal electrode sites in response to median nerve stimulation, reflects excitatory post-synaptic potentials (EPSPs) of the first thalamo-cortical volley (*Bruyns-Haylett et al., 2017*; *Peterson et al., 1995*; *Wikström et al., 1996*) which are generated in the anterior wall of the postcentral gyrus, Brodmann area 3b (*Allison et al., 1991*). Thus, the N20 directly reflects the intensity of a given stimulus. However, when keeping the sensory input constant, the amplitude of this early part of the SEP only depends on the excitability of the involved, well-defined

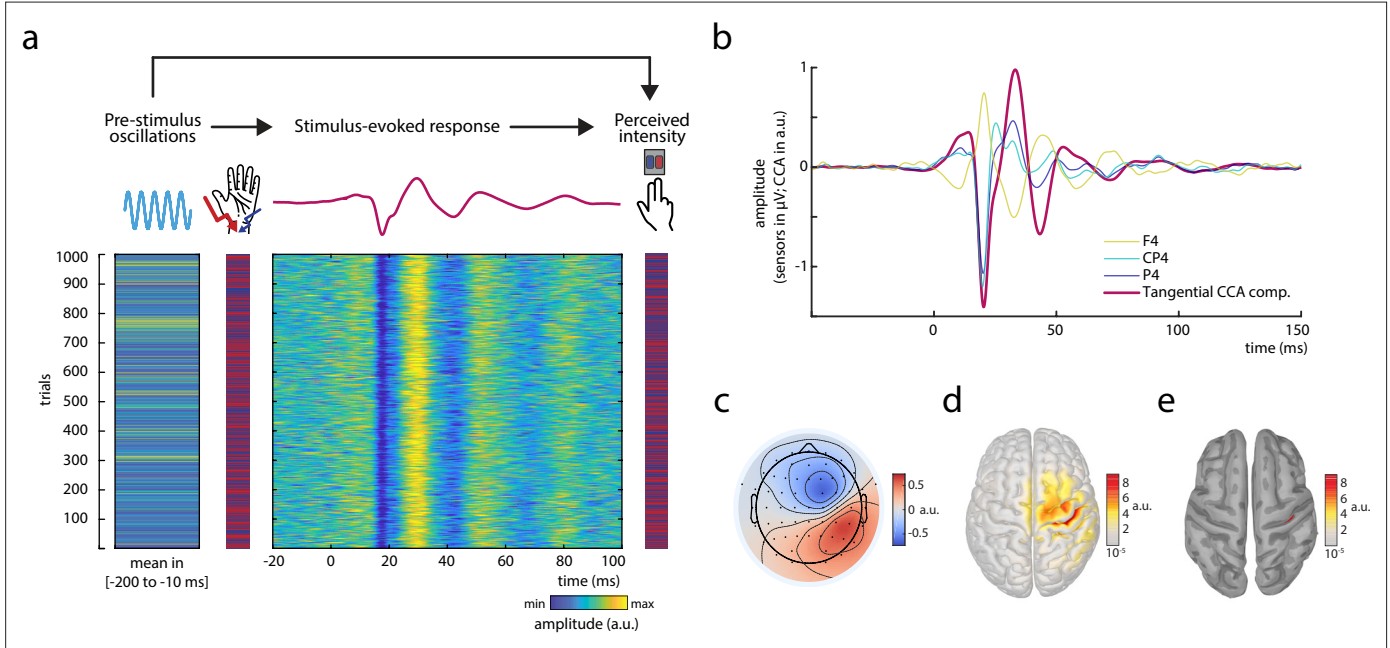

**Figure 1.** Experimental paradigm and main electrophysiological measures. (**a**) The relationships between pre-stimulus alpha oscillations, stimulus-evoked responses, and perceived intensity of somatosensory stimuli were examined in a continuous sequence of median nerve stimuli of two intensities with inter-stimulus intervals of ISI = 1513 ± 50 ms. After every stimulus, participants were to rate the perceived intensity as either 'strong' or 'weak' as fast as possible by button press. The raster plots represent the data of an exemplary subject with the rows corresponding to single trials. Displayed from left to right: Average pre-stimulus alpha amplitude, intensity of the presented stimuli (red = strong; blue = weak intensity), short-latency somatosensory evoked potentials (SEPs), and the perceived intensity as reported by the participants (red = strong; blue = weak intensity). Alpha activity and the SEP were both retrieved from the same tangentially oriented *canonical correlation analysis (CCA) component* (displayed in panels b–e) and hence reflect activity of the same neuronal sources. (**b**) Grand average of the SEP (*N* = 32) in sensor space (electrodes F4, CP4, and P4) and for the *tangential CCA component* as derived from the single-trial extraction approach using CCA. (**c**) Activation pattern of the *tangential CCA component* displaying a tangential dipole contralateral to stimulation site over the central sulcus which is typical for the N20-P35 complex of the SEP. Averaged across participants (*N* = 32). (**d**) Neuronal sources (absolute values) underlying the activation pattern of the *tangential CCA component*, reconstructed using eLoreta inverse modeling. Averaged across participants (*N* = 32). (**e**) Same as d but applying an amplitude threshold of 95 % in order to indicate the strongest generators of neural activity (displayed on a smoothed cortex surface).

neuronal population in the primary somatosensory cortex, and therefore represents an excellent instantaneous probe thereof. Notably, amplitude fluctuations of the N20 component have recently been found to relate to pre-stimulus alpha activity both at a given instance and through their long-term temporal dynamics, which suggests that both measures reflect a common modulating factor, that is, cortical excitability (*Stephani et al., 2020*).

In the current study, we set out to examine the implications of instantaneous excitability fluctuations at initial cortical processing – as measured by both pre-stimulus alpha activity and N20 amplitudes – on the perceived intensity of somatosensory stimuli. We used a binary intensity rating task in which participants were to discriminate supra-threshold median nerve stimuli of two intensities in a continuous stimulation sequence (*Figure 1a*). Our results show that both pre-stimulus alpha activity and N20 amplitudes are associated with a bias in the perceived intensity of somatosensory stimuli. Thus, instantaneous excitability fluctuations affect sensory brain responses already at earliest possible cortical processing with behaviorally relevant consequences. Counter-intuitively, elevated neural excitability and stronger stimulus intensity resulted in reverse effects on short-latency SEP amplitudes, which in turn may offer further insights into the neural mechanisms of excitability regulation through resting membrane potentials.

## Results

### Behavioral results

Participants discriminated the weak and the strong stimuli with an average accuracy of $acc_{mean}$ = 69.72% (SD = 7.94%; $CI_{95\%}$: [66.86%, 72.58%]), suggesting a moderate to high task difficulty. As confirmed by permutation tests, every individual participant performed better than chance level, all p < 0.05 (Bonferroni-corrected). The average discrimination sensitivity was $d'_{mean}$ = 1.14 (SD = 0.46; $CI_{95\%}$: [0.98, 1.31]) and the average criterion $c_{mean}$ = 0.01 (SD = 0.22; $CI_{95\%}$: [–0.066, 0.094]), according to signal detection theory (*Green and Swets, 1966*). Participants pressed the respective response button with an average reaction time of $RT_{mean}$ = 642.42 ms (SD = 95.79 ms; $CI_{95\%}$: [607.89 ms, 676.96 ms]).

### Extraction of single-trial SEPs

Single-trial activity of the early SEP was extracted using a variant of canonical correlation analysis (CCA; *Scheer et al., 2013*; *Waterstraat et al., 2015*), as previously reported for a similar paradigm examining the fluctuation of single-trial SEPs in response to stimuli with constant intensity (*Stephani et al., 2020*). This variant of CCA extracts a number of spatially distinct components based on a pattern matching between average SEP and single trials. Similar to *Stephani et al., 2020*, a prominent CCA component was identified in all subjects, which showed a clear peak at around 20 ms post-stimulus (*Figure 1b*) and displayed the pattern of the typical N20 tangential dipole (*Figure 1c*). Furthermore, neuronal sources of this CCA component were primarily located in the anterior wall of the post-central gyrus (Brodmann area 3b) in the primary somatosensory cortex (*Figure 1d and e*). Single-trial SEPs from this – as is referred to in the following – *tangential CCA component* are displayed for an exemplary subject in *Figure 1a*.

### Pre-stimulus alpha amplitude is associated with a bias in perceived stimulus intensity

To assess whether pre-stimulus neural states modulated the perception of upcoming somatosensory stimuli, we related oscillatory activity in the alpha band (8–13 Hz) before stimulus onset to the participants' reports of perceived stimulus intensity. Alpha band activity was measured from the same neural sources as the SEP, applying the spatial filters of the tangential CCA component. *Figure 2a* shows the envelope of pre-stimulus alpha activity depending on the behavioral responses of the participants. Pre-stimulus alpha amplitude was higher when participants rated the stimulus to be weak rather than strong (regardless of the actual stimulus intensity). This observation was further quantified with signal detection theory (SDT; *Green and Swets, 1966*) in order to differentiate the ability to discriminate stimulus intensities, as measured by *sensitivity d'*, from a response bias toward either strong or weak perceived intensity, as measured by *criterion c*. In correspondence with a recent study in the visual domain (*Iemi et al., 2017*), these SDT-derived parameters were statistically compared between the 20 % of trials with the lowest and the 20 % of trials with the highest pre-stimulus alpha amplitudes (as

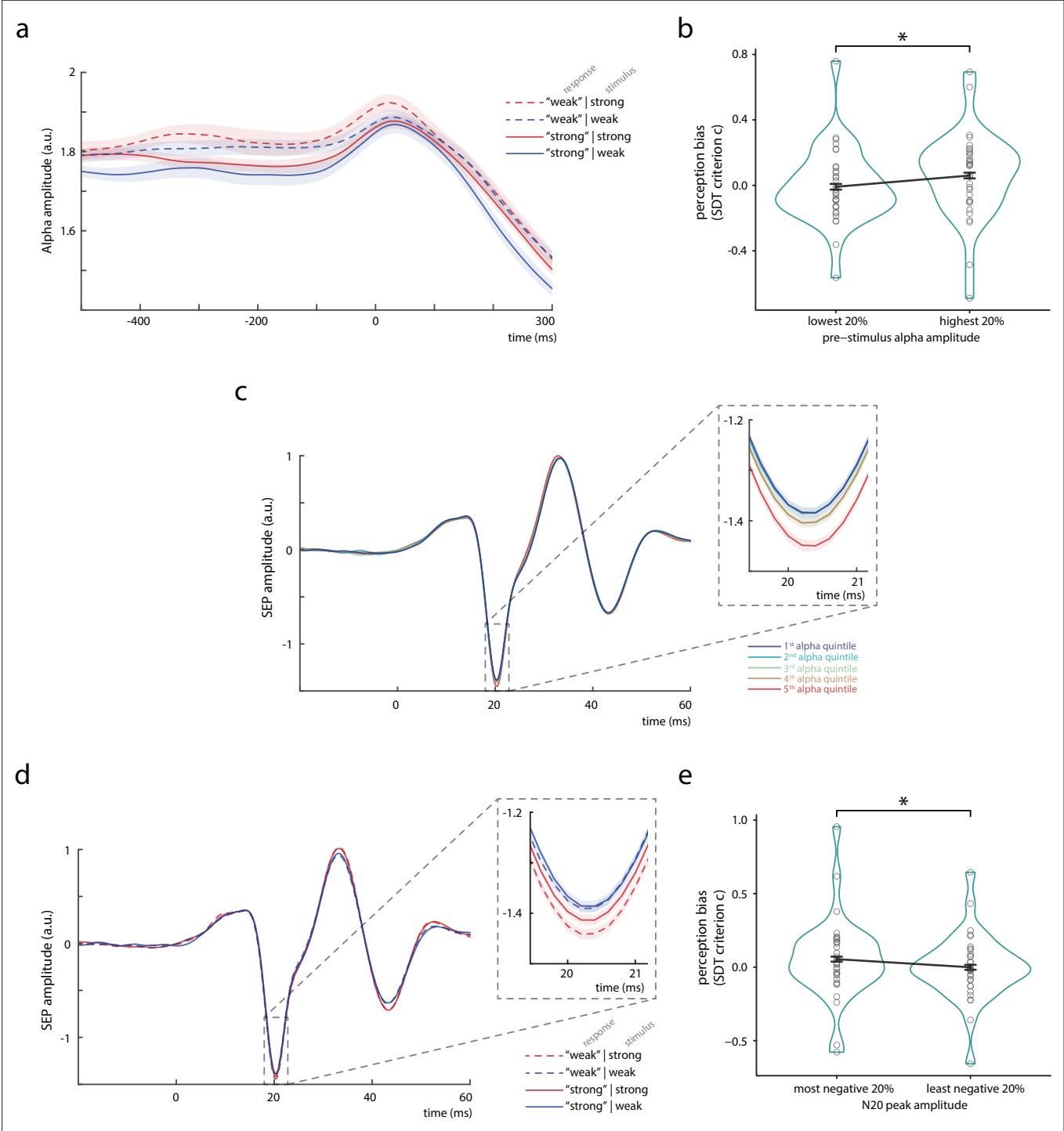

**Figure 2.** Bivariate relationships between pre-stimulus alpha amplitude, N20 peak amplitude, and perceived stimulus intensity. (**a**) Time course of the amplitude of pre-stimulus alpha band activity (8–13 Hz) displayed by behavioral response categories. Note that for statistical analyses, pre-stimulus epochs were cut at –5 ms relative to stimulus onset *before* filtering the data in the alpha band (8–13 Hz), in order to prevent contamination of the pre-stimulus window by stimulus-related activity. (**b**) Change in perception bias (i.e., signal detection theory [SDT] parameter *criterion c*) from the lowest to the highest alpha amplitude quintile (as measured between –200 and –10 ms). (**c**) Somatosensory evoked potential (SEP) derived from the tangential component of the canonical correlation analysis (CCA), sorted with respect to pre-stimulus alpha amplitude quintiles. Alpha quintiles were sorted in ascending order (i.e., first quintile = lowest alpha amplitude). (**d**) SEP (tangential CCA component) sorted according to behavioral response categories. (**e**) Change in perception bias (i.e., SDT parameter *criterion c*) from the most to the least negative N20 peak amplitude quintile. All panels show the grand average across all participants (*N* = 32). Shaded areas in panels a, c, and d, as well as error bars in panels b and e correspond to the standard

*Figure 2 continued on next page*

*Figure 2 continued*

errors of the mean based on the within-subject variances (*Morey, 2008*). Transparent circles in panels b and e reflect data of individual participants while black lines reflect the arithmetic mean on group level. Please refer to *Figure 2—figure supplement 1* for a schematic of SDT, to *Figure 2—source data 1* for details on the trial overlap between SDT analyses of pre-stimulus alpha activity and N20 amplitudes, to *Figure 2—figure supplement 2* for time-frequency representations of the observed effects, and to *Figure 2—figure supplement 3* for further control analyses using simulated SEP data.

The online version of this article includes the following source data and figure supplement(s) for figure 2:

**Source data 1.** Trial overlap between extreme bins of N20 and pre-stimulus alpha amplitudes used for the SDT analyses.

**Figure supplement 1.** Schematic of the signal detection theory parameters *sensitivity d'* and *criterion c*.

**Figure supplement 2.** Representation of the relations between amplitude in the time-frequency domain and N20 amplitudes (panel a), as well as perceived stimulus intensity (panel b).

**Figure supplement 3.** Simulation of filter effects on the relation between pre-stimulus alpha amplitude and early somatosensory evoked potential (SEP).

averaged in a time window from 200 to 10 ms before the stimulus onset; *Figure 2b*). A paired-sample *t*-test confirmed a difference regarding *criterion c*, $t(31) = -2.777$, p = 0.009, *Cohen's d* = $-0.491$, with average criterions of $c_{lowest20\%} = -0.009$ and $c_{highest20\%} = 0.060$ (CI$_{95\%}$ of difference: $[-0.119, -0.018]$). No difference was found for *sensitivity d'*, $t(31) = -1.425$, p = 0.164, *Cohen's d* = $-0.252$, with average sensitivities $d'_{lowest20\%} = 1.058$ and $d'_{highest20\%} = 1.142$ (CI$_{95\%}$ of difference: $[-0.204, 0.036]$). Thus, higher pre-stimulus alpha amplitude was associated with a higher threshold to rate the stimulus as 'strong', corresponding to a bias to generally report lower stimulus intensities, whereas the discriminability between the stimulus categories appeared unaffected.

## Pre-stimulus alpha amplitude is associated with single-trial N20 amplitudes

Following the hypothesis that both pre-stimulus alpha band activity and the N20 component of the SEP reflect changes in instantaneous cortical excitability, a covariation of these two measures should be expected (*Stephani et al., 2020*). Indeed, higher pre-stimulus alpha amplitudes were associated with larger (i.e., more negative) N20 peak amplitudes (*Figure 2c*), as statistically tested with a random-slope linear-mixed-effects model, $\beta_{fixed} = -0.023$, $t(32.17) = -2.969$, p = 0.006 (CI$_{95\%}$ of $\beta_{fixed}$: $[-0.040, -0.007]$). Notably, the direction of this effect may appear counter-intuitive at first sight but can be explained by the physiological basis of EEG generation, which offers important insights into the functional link between pre-stimulus alpha activity and SEP (see Discussion section *Opposing signatures of presented stimulus intensity and excitability in the early SEP*).

## Single-trial N20 amplitudes are associated with a bias in perceived stimulus intensity

Given the relationships of pre-stimulus alpha activity with perceived stimulus intensity and N20 peak amplitudes, we tested whether the latter also related to the SDT parameters of the behavioral performance. In parallel to the analyses of the effect of pre-stimulus alpha activity, *sensitivity d'* and *criterion c* were statistically compared between the 20 % of trials with the most negative and the 20 % of trials with the least negative N20 peak amplitudes. (Please note that this procedure resulted in a different trial selection as compared to the SDT analysis of pre-stimulus alpha activity. Please refer to *Figure 2—source data 1* for further details on the trial overlap.) Again, a significant difference was found for *criterion c*, $t(31) = 2.306$, p = 0.028, *Cohen's d* = 0.408, with average criterions of $c_{most\ neg.20\%} = 0.054$ and $c_{least\ neg.20\%} = -0.001$ (CI$_{95\%}$ of difference: $[0.006, 0.104]$), as assessed with a paired-sample *t*-test. No effect emerged for *sensitivity d'*, $t(31) = -1.747$, p = 0.091, *Cohen's d* = $-0.309$, with $d'_{leastneg.20\%} = 1.213$ and $d'_{mostneg.20\%} = 1.142$ (CI$_{95\%}$ of difference: $[-0.154, 0.012]$). Thus, c*riterion c* was lower for smaller than for larger N20 peak amplitudes (*Figure 2e*). This indicates that participants were more likely to rate a stimulus as 'strong' rather than 'weak' when the magnitude of the N20 potential was smaller (i.e., less negative), after taking into account the stimulus' actual intensity, as it also becomes evident from the SEPs sorted by the behavioral response categories (*Figure 2d*). Interestingly, the relationship between N20 amplitudes and perceptual outcome appeared to be driven mainly by differences within the strong stimulus category (*Figure 2d*). This may reflect the naturally higher signal-to-noise ratio (SNR)

of SEPs in response to stronger stimuli, or it could also point out that there was a 'floor effect' for the modulation of SEPs in response to the weaker stimuli.

## Structural equation modeling of effect paths

Importantly, *Figure 2d* also suggests that N20 amplitudes were generally larger (i.e., more negative) for higher stimulus intensities – thus showing an effect of opposite direction on N20 amplitudes as compared to instantaneous cortical excitability. In order to disentangle these effects of excitability and stimulus intensity, we examined their respective contributions in a two-level structural equation model, with *stimulus intensity*, *pre-stimulus alpha amplitude*, and *N20 peak amplitude* as predictors of *perceived stimulus intensity* on level 1 (within subjects), and random intercepts as well as their variances on level 2 (between subjects), including all single trials. Furthermore, we added the measures of compound nerve action potentials of the median nerve (*CNAP*; *Figure 3a and b*) and compound muscle action potentials of the M. abductor pollicis brevis (*CMAP*; *Figure 3c and d*) to the model, in order to control for peripheral variability.

On the one hand, both these peripheral measures should relate to stimulus intensity. On the other, there should be no effect of CNAP and CMAP on N20 amplitudes, when statistically controlling for stimulus intensity if the hypothesized fluctuations of excitability emerge on a cortical level. Yet, stimulus-induced thumb twitches may influence the participants' intensity ratings of the stimuli (even though the stimulated hand was covered with a paper box). The resulting two-level structural equation model (SEM 1; *Figure 4*) indicated statistical significance of all hypothesized effect paths, all $p_\beta \leq 0.003$, with model fit indices of Akaike information criterion (AIC) = 278,788.5, Bayesian information criterion (BIC) = 278,972.2, and log-likelihood = −139,372.2.

To evaluate the model fit, we compared a list of alternative models including or excluding relevant effect paths (*Table 1*). As indicated by chi-square difference tests, the log-likelihood of SEM 1 did not differ from those of SEMs 2–4, 9, and 10. Seeking model parsimony, SEM 1 is preferred over SEMs 2–4, 9, and 10 since the alternative models included one more parameter each, while fitting the data equally well. In comparison to SEMs 5–8, SEM 1 showed a significantly higher

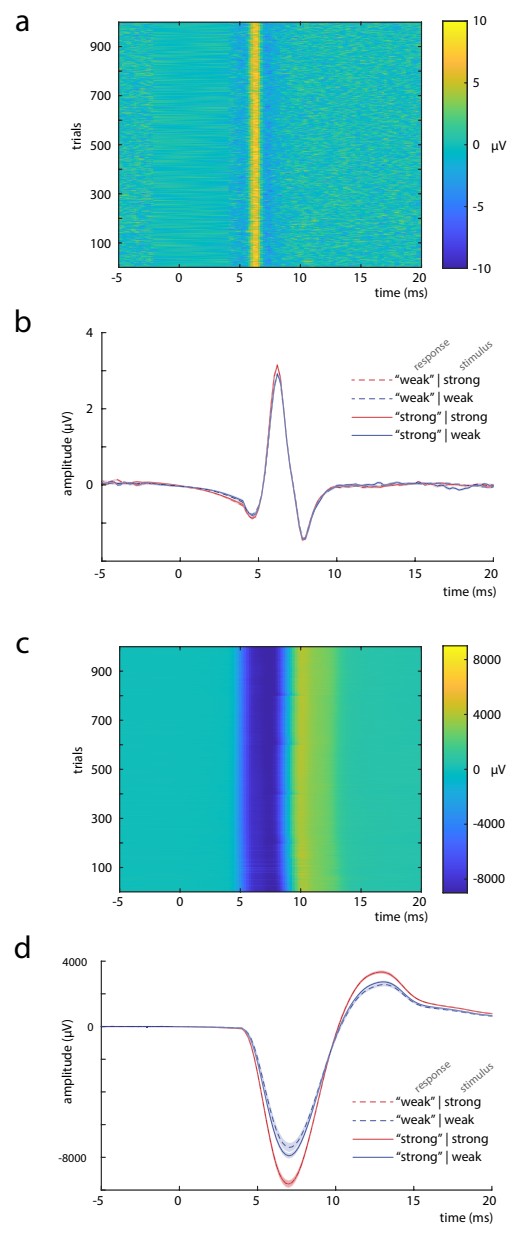

**Figure 3.** Measures to control for peripheral nerve variability. (**a**) Single trials of the *compound nerve action potential (CNAP)* in response to the median nerve stimuli, measured at the inner side of the ipsilateral upper arm (shown for an exemplary subject). (**b**) Grand average across participants (*N* = 32) of the CNAP, displayed by stimulus and response types. (**c**) Single trials of the *compound muscle action potential (CMAP)*, measured at the M. abductor pollicis brevis (shown for an exemplary subject). (**d**) Grand average across participants (*N* = 32) of the CMAP, displayed by stimulus and response types. Shaded areas in panels b and d correspond to the standard errors of the mean based on the within-subject variances (***Morey, 2008***).

log-likelihood suggesting a better model fit than these more parsimonious models. This is further

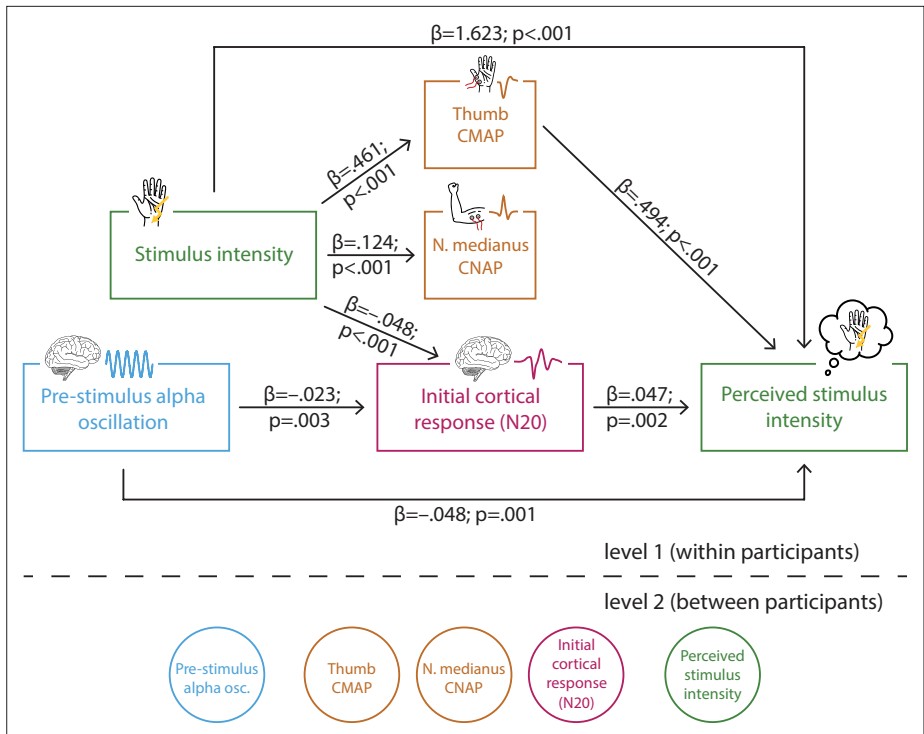

**Figure 4.** Multi-level structural equation model of the interplay between pre-stimulus alpha activity, the initial cortical response (N20 component of the somatosensory evoked potential [SEP]), intensity of the presented stimuli, the peripheral control measures of the compound muscle action potential (CMAP) of the M. abductor pollicis brevis and the compound nerve action potential (CNAP) of the median nerve, as well as the perceived intensity as reported by the participants (referred to as SEM 1). Effect paths were estimated between the manifest variables on level 1 (within participants). Latent variables on level 2 served to estimate the respective random intercepts as well as their between-subject variances according to the latent variable approach for multi-level models as implemented in *Mplus*.

supported by the AIC and BIC values which were altogether lowest for SEM 1. Hence, we conclude that SEM 1 fitted our empirical data best.

The estimated path coefficients (*Figure 4*) correspond well with above reported bivariate relationships: When controlling for stimulus intensity, both higher pre-stimulus alpha amplitudes and larger (i.e., more negative) N20 amplitudes were associated with a lower perceived intensity (equivalent to a response bias as reflected in *criterion c*), as well as higher pre-stimulus alpha amplitudes co-occurred with larger (i.e., more negative) N20 amplitudes. In addition, the SEM further dissociated the effects of stimulus intensity on early electrophysiological measures and their respective effects on perceived stimulus intensity. Higher stimulus intensity was associated with larger N20 amplitudes, which constitutes an effect of opposite direction as compared to the N20-related excitability effect on perceived intensity. Furthermore, higher stimulus intensity also led to larger amplitudes of CMAP and CNAP, due to the physical difference in stimulation strength, as could be expected a priori. Additionally, larger CMAP amplitudes resulted in a higher perceived intensity, while no such effect was observed for CNAP. Importantly, neither CMAP nor CNAP related to N20 amplitudes when controlling for stimulus intensity. Thus, fluctuations in cortical processing were not driven by peripheral variability. Also, peripheral activity in CMAP and CNAP did not show any association with pre-stimulus alpha activity. Finally, a substantial effect on the perceived intensity was found for stimulus intensity. This was expected as the overall accuracy in the discrimination task was about 70 %.

Taken together, the SEM confirms the hypothesized influences of instantaneous fluctuations of early SEPs as well as pre-stimulus oscillatory activity on the consciously accessible percept of a stimulus. Moreover, this analysis demonstrates that stimulus intensity and cortical excitability, which in turn determines the perceived stimulus intensity, show opposing effects on the amplitude of the early SEP.

**Table 1.** Model comparison of structural equation models (SEMs).

The original SEM (1) was compared to the alternative models (2–10) using Akaike information criterion (AIC), Bayesian information criterion (BIC), log-likelihood (LL), and the chi-square difference test based on the LL (with corresponding p-value; p-values < 0.05 are indicated by bold print). Differences in AIC, BIC, LL, and degrees of freedom (df) were derived by the subtraction *alternative SEM minus SEM 1*. A better model fit is indicated by lower AIC and/or BIC as well as higher LL. The $\chi^2$ difference tests correspond to the comparisons *model with fewer parameters minus model with more parameters*.

| | Model fit indices | | | | | |
| --- | --- | --- | --- | --- | --- | --- |
| | AIC diff. | BIC diff. | LL diff. | $\chi^2$ diff. | df diff. | p-value |
| (1) Original SEM ('SEM 1') | | | | | | |
| (2) SEM incl. N20 ~ CNAP | 1.813 | 10.166 | 0.093 | 0.146 | −1 | 0.702 |
| (3) SEM incl. N20 ~ CMAP | 0.088 | 8.441 | 0.956 | 0.799 | −1 | 0.371 |
| (4) SEM incl. perceived_int ~ CNAP | 1.967 | 10.320 | 0.016 | 0.019 | −1 | 0.890 |
| (5) SEM excl. perceived_int ~ prestim | 8.002 | −0.351 | −5.001 | 11.415 | 1 | **<0.001** |
| (6) SEM excl. N20 ~ prestim | 15.053 | 6.701 | −8.527 | 8.087 | 1 | **0.005** |
| (7) SEM excl. N20 | 47.099 | 22.040 | −26.550 | 31.095 | 3 | **<0.001** |
| (8) SEM excl. CMAP | 9586.906 | 9570.200 | −4795.453 | 87.030 | 2 | **<0.001** |
| (9) SEM incl. CMAP ~ prestim | 1.404 | 9.757 | 0.297 | 0.220 | −1 | 0.639 |
| (10) SEM incl. CNAP ~ prestim | −0.115 | 8.239 | 1.057 | 2.342 | −1 | 0.126 |

## Reconstruction of the observed effects in source space

In order to investigate whether the observed effects of pre-stimulus alpha activity on N20 amplitudes and the perceived stimulus intensity were specific only to the generator regions of the SEP, we repeated the SDT analysis as well as the linear-mixed-effects models for these relations in source space (i.e., separately for every source estimated based on individual head models; see Materials and methods). As visible from *Figure 5*, the effects of pre-stimulus alpha amplitude on both N20 amplitude and perceived stimulus intensity were indeed most pronounced around the hand region of the right primary somatosensory cortex, the same region which we identified as source for the tangential CCA component used in the analyses above. The effects in source space using the SDT approach did not reach significance after the correction for multiple comparisons. Yet, a region of interest (ROI) analysis within the hand region of the right primary somatosensory cortex did confirm the observations from our previous analyses: There was an effect of pre-stimulus alpha amplitude on SDT parameter *criterion c*, $t(31) = -2.951$, $p = 0.006$, $CI_{95\%} = [-0.173, -0.032]$, but no effect on *sensitivity d'*, $t(31) = 0.633$, $p = 0.531$, $CI_{95\%} = [-0.083, 0.157]$. (Please refer to *Figure 5—figure supplement 1* for the distribution of the SDT effects across the whole cortex.) Taken together, the relationships between pre-stimulus alpha activity, N20 potential of the SEP, and perceived stimulus intensity appear to be attributable to neural activity from the same (or at least very similar) sources in the right primary somatosensory cortex.

## Variability in thalamus-related activity is not related to behavioral responses

To examine further whether the observed neuronal effects on the perceived stimulus intensity were of a cortical origin, we analyzed the EEG responses prior to the N20 potential. In a sub-sample of 13 participants, the CCA decomposition provided a component that showed a clear peak at 15 ms, characterized by a spatial pattern that suggested a deep, medial source (*Figure 6a*). Most likely, this CCA component thus corresponds to the P15 potential of the SEP, which is thought to reflect activity in the thalamus (*Albe-Fessard et al., 1986*). The amplitude of this P15 component did not relate to the perceived stimulus intensity, as examined with a random-intercept linear-mixed-effects model with

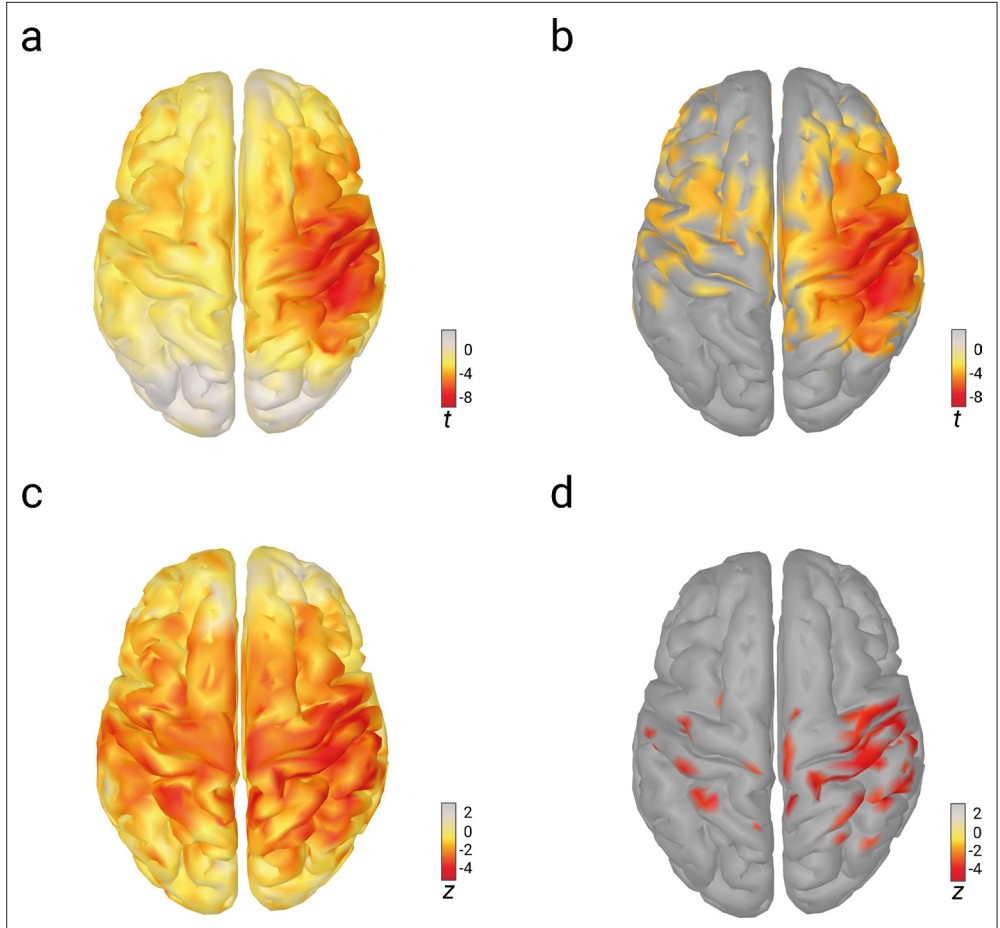

**Figure 5.** Relations of pre-stimulus alpha amplitude with N20 amplitudes and perceived stimulus intensity as analyzed with linear-mixed-effects models in source space. (**a**) Effects of pre-stimulus alpha amplitudes on N20 amplitudes (uncorrected *t* values). (**b**) Same as (**a**) but corrected for multiple comparisons (FDR-corrected; p < 0.01). (**c**) Effects of pre-stimulus alpha amplitudes on perceived stimulus intensity (uncorrected *z* values). (**d**) Same as (**c**) but corrected for multiple comparisons (FDR-corrected; p < 0.01).

The online version of this article includes the following figure supplement(s) for figure 5:

**Figure supplement 1.** Effects of pre-stimulus alpha amplitude on signal detection theory parameters *criterion c* and *sensitivity d'* in source space.

---

*perceived intensity* as dependent variable and *P15 amplitude* and *stimulus intensity* as predictors, $\beta_{P15}$ = 0.008, z = 0.394, p = 0.694 (CI$_{95\%}$ of $\beta_{P15}$: [–0.031, 0.047]). As expected, *stimulus intensity* however showed a significant effect on *perceived intensity*, $\beta_{stim\_int}$ = 1.851, z = 46.463, p < 0.001 (CI$_{95\%}$ of $\beta_{stim\_int}$: [1.773, 1.929]). Additionally, we calculated the statistical power of finding an effect of P15 amplitude in the linear-mixed-effects model, using Monte Carlo simulations (*Green et al., 2016*) and assuming an effect size comparable to the observed N20 effect on perceived intensity. The post hoc power analysis revealed a statistical power of 71.9 %. In addition, the SDT analysis based on binning of the P15 amplitudes into quintiles neither suggested a relation with *criterion c* nor *sensitivity d'*, t(12) = 1.201, p = 0.253, and t(12) = –0.201, p = 0.844, respectively. Therefore, we conclude that it is unlikely that the effect of N20 amplitudes on perceived stimulus intensity was driven by thalamic variability and that the modulation of perceived stimulus intensity emerges rather on the cortical level, reflecting instantaneous changes of cortical excitability. However, P15 amplitudes did also not differ between different presented stimulus intensities, as tested with a random-intercept linear-mixed-effects model, $\beta_{stim\_int}$ = 0.009, t(12721.82) = 0.531, p = 0.596 (CI$_{95\%}$ of $\beta_{stim\_int}$: [–0.025, 0.043]), which may indicate that this EEG-based measure of thalamic activity is generally not very sensitive to differences between experimental conditions. For completeness, we also tested for the effect of pre-stimulus alpha amplitude on

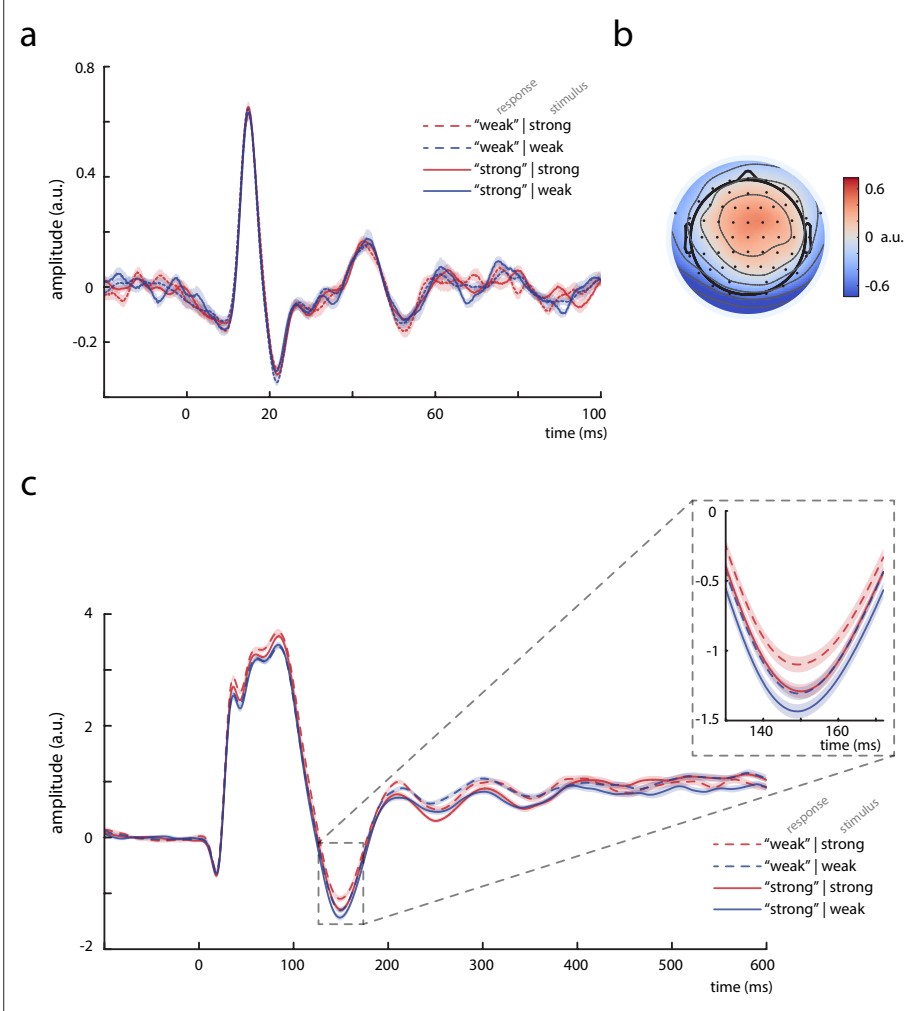

**Figure 6.** Thalamic activity and later somatosensory evoked potential (SEP) components. (**a**) Grand average (*N* = 13) of the thalamic component derived from canonical correlation analysis (CCA), showing a clear P15 potential which did not differ across behavioral response categories. (**b**) Activation pattern of the thalamic CCA component (average across subjects). (**c**) Grand average (*N* = 32) of later SEP components (extracted with the tangential-CCA filter in the frequency range from 0.5 to 45 Hz). The N140 is visible as a negative peak at around 149 ms. Larger N140 amplitudes are associated with higher perceived intensities. Shaded areas correspond to the standard errors of the mean based on the within-subject variances (*Morey, 2008*).

P15 amplitude, which was also not significant, $\beta_{prestim}$ = –0.007, *t*(12210.50) = –0.635, p = 0.526 (CI$_{95\%}$ of $\beta_{prestim}$: [–0.028, 0.014]).

## Effects in later SEP components conform with previous studies

In order to relate our novel findings in early SEPs to the existing literature on somatosensory processing at later stages, we additionally examined a well-studied later component of the SEP, the N140. For this SEP component, a larger amplitude has typically been associated with a stronger percept of the presented stimulus (e.g., *Al et al., 2020*; *Schröder et al., 2021*; *Schubert et al., 2006*). Indeed, a comparable effect of N140 amplitude on perceived intensity was also present in our data (*Figure 6c*), as statistically tested with a random-slope linear-mixed-effects model, $\beta_{fixed}$ = –0.058, *z* = –3.387, p < 0.001 (CI$_{95\%}$ of $\beta_{fixed}$: [–0.093, –0.024]). Also, the presented stimulus intensity was related to the perceived intensity, $\beta_{fixed}$ = 1.876, *z* = 14.015, p < 0.001 (CI$_{95\%}$ of $\beta_{fixed}$: [1.605, 2.145]), which was expected given the participants' discrimination performance being above chance level. These findings were in line with a separate SDT analysis: N140 amplitudes were associated with an effect on *criterion c*, *t*(31) = –3.010, p = 0.005, but no effect on *sensitivity d'* emerged, *t*(31) = 0.246, p = 0.807.

In addition, a second random-slope linear-mixed-effects model indicated that N140 amplitudes were in turn modulated by the presented stimulus intensity, $\beta_{fixed}$ = 0.030, $t$(30.92) = 2.275, p = 0.030 ($CI_{95\%}$ of $\beta_{fixed}$: [0.004, 0.056]), as well as by pre-stimulus alpha activity, $\beta_{fixed}$ = 0.021, $t$(31.70) = 2.73, p = 0.030 ($CI_{95\%}$ of $\beta_{fixed}$: [0.002, 0.040]).

Taken together, our results are thus consistent with previous studies on the relation between pre-stimulus state and somatosensory processing at later stages, while demonstrating opposing effects for the very first cortical response.

## Discussion

Using a somatosensory discrimination paradigm, we examined the modulation of perceived stimulus intensity by instantaneous fluctuations of cortical excitability at initial cortical processing. Both pre-stimulus alpha band activity and initial cortical evoked responses were associated with a bias in intensity discrimination, suggesting that a lower cortical excitability reduces the perceived intensity of sensory stimuli. Furthermore, we rule out that variability in peripheral nerve activity accounted for these effects, in line with the notion of instantaneous excitability changes being intrinsic to cortical brain dynamics. Intriguingly, elevated excitability and higher presented stimulus intensity resulted in opposing amplitude effects on the initial stimulus-related response in the cortex, the N20 component of the SEP. Based on the neurophysiological principles of the EEG generation, this finding may be explained by a mechanistic link between pre-stimulus alpha activity and initial cortical EPSPs through modulations of resting membrane potentials.

### Fluctuations of cortical excitability affect the perceived stimulus intensity

In line with previous studies on the modulatory role of alpha oscillations on perceptual processes (*Craddock et al., 2017*; *Iemi et al., 2017*), we found higher pre-stimulus alpha amplitudes to be associated with a lower perceived intensity of somatosensory stimuli. This was indicated both by an increased threshold (*criterion c*) of reporting a higher stimulus intensity according to SDT (*Green and Swets, 1966*) and by the negative relationship between pre-stimulus alpha amplitude and reported stimulus intensity in the structural equation model. Moreover, sensory processing appeared to be modulated by ongoing oscillatory activity already during initial cortical responses, as suggested by the relations between pre-stimulus alpha activity and N20 amplitude, as well as N20 amplitude and perceived stimulus intensity. The N20 component of the SEP reflects initial stimulus-related excitatory activity (i.e., EPSPs) resulting from the first thalamo-cortical volley to the primary somatosensory cortex (*Bruyns-Haylett et al., 2017*; *Peterson et al., 1995*; *Wikström et al., 1996*) and thus represents a direct measure of cortical excitability (when keeping the stimulus intensity constant). The modulation of perceived stimulus intensity therefore relates to a sensory bias at earliest possible cortical processing, reflecting fluctuations of instantaneous neural excitability.

Furthermore, these findings demonstrate that effects of pre-stimulus oscillatory activity on the processing of sensory stimuli are not restricted to near-threshold stimuli where a detection threshold is assumed to be shifted by ongoing brain activity (*Iemi et al., 2017*; *Samaha et al., 2020*). Instead, our findings suggest that cortical excitability can affect the representation of stimulus features in supra-threshold perception, too. Importantly, our criterion-free discrimination paradigm of two neutral response alternatives ('strong' or 'weak') precluded the potential confounding effect of perceptual confidence, which has recently been considered as an alternative explanation for pre-stimulus alpha effects on perceptual biases (*Benwell et al., 2017*; *Samaha et al., 2017*). In our forced-choice paradigm, different levels of perceptual confidence could not have influenced the intensity ratings since the task was to distinguish two clearly perceptible stimuli, and not to report whether a stimulus was perceived or not (as done in near-threshold paradigms). Thus, the current findings unequivocally indicate – to the best of our knowledge for the first time – that pre-stimulus alpha oscillations affect the behavioral outcome via a modulation of the internally represented stimulus intensity.

## Opposing signatures of presented stimulus intensity and excitability in the early SEP

Following the hypothesis of higher alpha activity being associated with lower cortical excitability (*Jensen and Mazaheri, 2010*; *Klimesch et al., 2007*; *Samaha et al., 2020*), it may seem counter-intuitive that higher alpha amplitudes were associated with larger (i.e., more negative) N20 amplitudes in our data. However, as we proposed recently (*Stephani et al., 2020*), this relationship may be explained by the neurophysiological mechanisms of EEG generation. The generated voltage on the scalp, $U$, in our case relating to the N20 potential, can be defined in the following way (*Ilmoniemi and Sarvas, 2019*; *Kandel et al., 2000*; *Lopes da Silva, 2004*):

$$U \sim I * N_{neurons} * LF,$$

where $I$ denotes the sum of local primary post-synaptic currents due to the activation of a given neuron, $N_{neurons}$ the number of involved neurons, and $LF$ the lead field coefficient projecting source activity to the electrodes on the scalp. Since the spatial arrangement of the neural generators and the EEG sensors was stable across stimulation events, $LF$ reflects a constant in the measurement of the N20 potential. In contrast, $N_{neurons}$ should increase with stimulus intensity since more nerve fibers are excited at stimulation site when applying stimuli of higher currents. This should lead to an increase of SEP amplitude with stimulus intensity, as reported in previous studies (*Jousmäki and Forss, 1998*; *Klostermann et al., 1998*) and as was observed in the current dataset for cortical (*Figure 2d*) as well as peripheral responses (*Figure 3b and d*). For constant stimulus intensity, however, $N_{neurons}$ is expected to stay approximately constant and amplitudes in the EEG should primarily depend on $I$, reflecting excitatory post-synaptic currents (EPSCs) in case of the N20 component. Crucially, EPSCs directly depend on the electro-chemical driving forces produced by the membrane potential. When moving the membrane potential toward depolarization – a state of higher excitability – the electro-chemical driving force for further depolarizing inward trans-membrane currents is decreased (*Castro-Alamancos, 2009*), which leads to smaller EPSCs (*Deisz et al., 1991*), and should in turn result in smaller amplitudes of the scalp EEG. Assuming an inverse relationship between the amplitude of alpha oscillations and neuronal excitability (as for example indicated by a lower neural firing rate during higher alpha activity; *Haegens et al., 2011*), one should hence rather expect *decreased* N20 amplitudes following *low* pre-stimulus alpha activity. This is what was observed in our data when controlling for stimulus intensity (*Figure 4*). Moreover, the notion of smaller (i.e., less negative) N20 amplitudes reflecting a state of higher excitability is corroborated by the behavioral data: When controlling for stimulus intensity, we found smaller N20 amplitudes to be associated with higher perceived stimulus intensity. Notably, the attenuation of early SEPs during high excitability states is in fact in line with previous observations from biophysically realistic modeling of oscillatory activity in the somatosensory cortex as well as its interplay with stimulus-evoked responses (*Jones et al., 2009*).

Taken together, our findings thus demonstrate that the intensity of the presented stimulus and the degree of instantaneous neural excitability are jointly reflected in the early SEP but with opposing signatures: While stronger stimulus intensity increases the N20 potential, *decreased* N20 amplitudes appear to be associated with an *increase* in excitability (which in turn lead to a higher perceived stimulus intensity). This challenges previous assumptions that the amplitude of brain potentials, especially at early processing stages, reflects the coding of the perceived stimulus intensity. Rather, our findings call for a more differentiated view. Although the amplitude of early event-related potentials may indeed reflect the size of the input (e.g., a stronger or weaker somatosensory stimulus), the neural evaluation of this input (i.e., the perceived intensity), however, further depends on internal neural states, such as neural excitability, which may even reverse the amplitude effects of the input already at the earliest cortical processing stages. Crucially, our data are at the same time consistent with previous studies on somatosensory processing at later stages, where larger EEG potentials are typically associated with a stronger percept of a given stimulus (e.g., *Al et al., 2020*; *Schröder et al., 2021*; *Schubert et al., 2006*), as both our SDT and linear-mixed-effects analyses of the N140 component showed. Thus, the present findings of opposing signatures of neural excitability and sensory input appear to be a distinct characteristic of early cortical potentials, involving the first bottom-up sensory processing. In this context, it should be further emphasized that our proposed physiological model may be directly applicable to well-isolated neural signals only (such as the N20 component of the SEP). The physiological interpretation of amplitudes of later EEG potentials, such as the N140,

however, is not as straightforward as described above, since several distinct SEP components may interact (*Auksztulewicz et al., 2012*), and excitatory and inhibitory contributions cannot be readily distinguished. Furthermore, with the present data, we cannot unambiguously conclude that the observed relation between pre-stimulus alpha activity and initial SEP indeed involved the very same neuronal populations – which may represent a limitation of the hypothesized mechanism. However, all approaches to localize these effects pointed to very similar cortical regions (as discussed in the following section). Also, we would like to emphasize that the presented mechanism reflects a hypothesized model, which shall be further supported or falsified with more targeted studies, for example, directly quantifying membrane potentials and trans-membrane currents in relation to different excitability states in somatosensation.

## Origin of excitability fluctuations

To further narrow down the neuronal sources that eventually led to fluctuations of the perceptual outcome, we controlled for peripheral nerve variability, extracted spatially well-defined EEG potentials (i.e., as reflected in the *tangential components* of the single-trial CCA), re-analyzed the effects of interest in source space, and examined subcortical activity.

Variability in afferent peripheral activity, as measured by CNAP at the upper arm, did not influence the perceived stimulus intensity when controlling for stimulus intensity. However, a robust effect on the perceived stimulus intensity was observed for efferent peripheral activity, as measured by CMAP of the M. abductor pollicis brevis. This may be explained by differences in proprioceptive sensations associated with the thumb twitches elicited by the stimulation, whose extent may depend on changes of the prevailing muscle tonus. Importantly, neither the CNAP nor the CMAP related to cortical excitability as measured by the N20 component. Thus, the excitability effects in the early SEP are distinct from variability in peripheral nerve activity.

Furthermore, pre-stimulus alpha band activity and the N20 component of the SEP were retrieved from the same neuronal sources, which – as indicated by source reconstruction of the *tangential CCA components* – were localized in the primary sensory cortex, centered around the hand region of Brodmann area 3b. An additional, SEP-independent analysis of the effects of pre-stimulus alpha amplitude on N20 amplitude and perceived stimulus intensity confirmed these findings in source space. Although one should bear in mind the limited spatial resolution of EEG, this further supports the notion of excitability fluctuations in primary sensory regions of the cortex, being reflected in both ongoing and evoked neural activity.

Another possibility is that already subcortical sources – particularly in the thalamus – may play a role in modulating sensory excitability and hence shape the perceptual outcome (*Kosciessa et al., 2021*). Yet, neither our SDT analyses nor the linear-mixed-effects models of the thalamus-related P15 component supported this notion. Although we estimated an acceptable statistical power of these analyses, it should be noted that we could also not observe an effect of presented stimulus intensity on P15 amplitudes. Therefore, these results should be interpreted with some caution as this measure may lack the required SNR to detect rather subtle experimental effects of interest such as of the intensity difference of two very similar stimuli. Nevertheless – and also taking into account our analyses in source space – we conclude that the findings of the present study are most consistent with the idea that the modulation of perceived intensity had its origins at the cortical level.

However, it remains an open question whether the observed excitability changes reflect local or global neural dynamics. Although there is initial evidence that cortical excitability may be organized temporally in a scale-free manner (*Stephani et al., 2020*), which may reflect an embedding into global critical-state dynamics (*Avramiea et al., 2020*; *Beggs and Plenz, 2003*; *Palva et al., 2013*), future work has to examine the spatial organization of excitability more specifically across different somatotopic projections in primary sensory areas as well as across diverse brain regions.

In addition, it is unclear at this point how exactly the observed fluctuations of initial cortical responses are integrated in later, downstream neural processes. In principle, changes in early sensory processing should provide the ground for later neural activity involved in the perceptual decision making, and finally shape the behavioral outcome (as observed in the current study). However, with our data, we cannot unambiguously tell whether the modulation of alpha oscillations – associated with excitability changes at the earliest cortical level – may in turn reflect a top-down regulated signal, which could enable the neural system to account for ongoing fluctuations of excitability and even

benefit from a certain degree of variability. Although more and more evidence indeed suggest an adaptive, functional role of neural 'noise' (e.g., **Findling and Wyart, 2021**), further studies are needed to better understand how this concept may pertain also to such fundamental neural properties as the initial cortical excitability to external sensory stimuli.

## Conclusions

Both ongoing oscillatory alpha activity and amplitude fluctuations of the first cortical response shape the perceived intensity of somatosensory stimuli. These effects most likely reflect instantaneous changes of cortical excitability in the primary somatosensory regions of the cortex, leading to a sensory bias which manifests already during the very first cortical response. Further questioning previous assumptions of how the evaluation of stimulus intensity is reflected in brain potentials, cortical excitability and the presented stimulus intensity were associated with opposing effects on the early SEP. We argue that this disparity may be explained by a mechanistic link between ongoing oscillations and stimulus-evoked activity through membrane potential alterations. This sheds new light on the neural correlates of the intensity encoding of somatosensory stimuli, which may well apply to other sensory domains, too.

# Materials and methods

## Participants

A total of 32 participants (all male, mean age = 27.0 years, SD = 5.0) were recruited from the database of the Max Planck Institute for Human Cognitive and Brain Sciences, Leipzig, Germany. As assessed using the Edinburgh Handedness Inventory (**Oldfield, 1971**), all participants were right-handed (lateralization score, $M$ = + 93.1, SD = 11.6). No participant reported any neurological or psychiatric disease. All participants gave informed consent and were reimbursed monetarily. The study was approved by the local ethics committee (Ethical Committee at the Medical Faculty of Leipzig University, Leipzig, Germany). The chosen sample size was based on previous, similar studies which yielded an appropriate statistical power. (Please also note here that the estimates of the within-subject effects analyzed in the current study were based on the analysis of single trials [nearly 1000 per participant], ensuring a sufficiently high statistical power.)

## Stimuli

Somatosensory stimuli were applied using electrical stimulation of the median nerve. A non-invasive bipolar stimulation electrode was positioned on the left wrist (cathode proximal). The electrical stimuli were designed as squared pulses of a 200 µs duration and applied using a DS-7 constant-current stimulator (Digitimer, Hertfordshire, United Kingdom). Stimuli of two intensities were presented, in the following referred to as *weak* and *strong* stimulus. The intensity of the weak stimulus was set to 1.2 times the motor threshold, leading to a clearly visible thumb twitch for every stimulus. The individual motor threshold was determined as the lowest intensity for which a thumb twitch was visible to the experimenter, as determined by a staircase procedure. The intensity of the strong stimulus was adjusted during training blocks prior to the experiment so that it was barely above the *just-noticeable difference*, corresponding to a discrimination sensitivity of about $d'$ = 1.5 according to Signal Detection Theory (SDT; **Green and Swets, 1966**). Thus, the stimulation intensities of the two stimuli were only barely distinguishable (despite both being clearly perceivable), with average intensities of 6.60 mA (SD = 1.62) and 7.93 mA (SD = 2.06), for the weak and strong stimulus, respectively.

## Procedure

During the experiment, participants were seated comfortably in a chair their hands extended in front of them in the supinate position on a pillow. The left hand and wrist, to which the stimulation electrodes were attached, was covered with a cardboard box in order to prevent the participants to judge the stimulus intensity visually by the extent of thumb twitches elicited by the stimulation. Weak and strong stimuli were presented with an equal probability in a continuous, pseudo-randomized sequence with inter-stimulus intervals (ISI) ranging from 1463 to 1563 ms (randomly drawn from a uniform distribution; $ISI_{average}$ = 1513 ms). In total, 1000 stimuli were applied, divided into five blocks of 200 stimuli each with short breaks in between. Participants were to indicate after each stimulus whether it was the

weak or strong stimulus, by button press with their right index and middle fingers as fast as possible. The button assignment for weak and strong stimulus was balanced across participants. Furthermore, every sequence started with a weak stimulus in order to provide an anchor point for the intensity judgments (participants were informed about this). While performing the discrimination task, participants were instructed to fixate their gaze on a cross on a computer screen in front of them.

Prior to the experiment, training blocks of 15 stimuli each were run in order to familiarize the participants with the task and to individually adjust the intensity of the strong stimulus so that a discrimination sensitivity of about $d' = 1.5$ resulted (the intensity of the weak stimulus was set at 1.2 times the motor threshold for all participants). On average across participants, this procedure comprised 10.5 training blocks (SD = 5.8). During these training blocks, participants were provided with visual feedback of their response accuracy. No information on task performance was given during the experimental blocks.

## Data acquisition

EEG data were recorded from 60 Ag/AgCl electrodes at a sampling rate of 5000 Hz using an 80-channel EEG system (NeurOne Tesla, Bittium, Oulu, Finland). A built-in band-pass filter in the frequency range from 0.16 to 1250 Hz was used. Electrodes were mounted in an elastic cap (EasyCap, Herrsching, Germany) at the international 10–10 system positions FP1, FPz, FP2, AF7, AF3, AFz, AF4, AF8, F7, F5, F3, F1, Fz, F2, F4, F6, F8, FT9, FT7, FT8, FT10, FC5, FC3, FC1, FC2, FC4, FC6, C5, C3, C1, Cz, C2, C4, C6, CP5, CP3, CP1, CPz, CP2, CP4, CP6, T7, T8, TP7, TP8, P7, P5, P3, P1, Pz, P2, P4, P6, P8, PO7, PO3, PO4, PO8, O1, and O2, with FCz as the reference and POz as the ground. For the purpose of source reconstruction, the electrode positions were measured in 3D space individually for each subject using the Polhemus Patriot motion tracker (Polhemus, Colchester, Vermont). In order to record the electrooculogram, four additional electrodes were positioned at the outer canthus and the infraorbital ridge of each eye. The impedances of all electrodes were kept below 10 kΩ. For source reconstruction, EEG electrode positions were measured in 3D space individually for each subject using Polhemus Patriot (Polhemus, Colchester, Vermont). Additionally, the compound nerve action potential (CNAP) of the median nerve and the compound muscle action potential (CMAP) of the M. abductor pollicis brevis were measured. For the CNAP, two bipolar electrodes were positioned on the inner side of the left upper arm along the path of the median nerve, at a distance of about 1 cm (reference electrode distal). The CMAP was measured from two bipolar electrodes placed on the stimulated hand, one on the muscle belly of the M. abductor pollicis brevis and the other on the second joint of the thumb (reference electrode).

Structural T1-weighted MRI scans (MPRAGE) of all participants, but two were obtained from the database of the Max Planck Institute for Human Cognitive and Brain Sciences, Leipzig, Germany, acquired within the same year of the experiment or up to 3 years earlier on a 3T Siemens Verio, Siemens Skyra, or Siemens Prisma scanner (Siemens, Erlangen, Germany).

## EEG pre-processing

Stimulation artifacts were cut out and interpolated between –2 and 4 ms relative to stimulus onset using Piecewise Cubic Hermite Interpolating Polynomials (MATLAB function *pchip*). The EEG data were band-pass filtered between 30 and 200 Hz, sliding a fourth-order Butterworth filter forward and backward over the data to prevent phase shift (MATLAB function *filtfilt*). As outlined in a previous study (*Stephani et al., 2020*), this filter allowed to specifically focus on the N20-P35 complex of the SEP, which emerges from frequencies above 35 Hz, and to omit contributions of later (slower) SEP potentials of no interest. Additionally, this filter effectively served as baseline correction of the SEP since it removed slow trends in the data, reaching an attenuation of 30 dB at 14 Hz, thus ensuring that fluctuations in the SEP did not arise from fluctuations within slower frequencies (e.g., alpha band activity). Subsequently, segments of the data that were distorted by muscle or non-biological artifacts were removed by visual inspection. After re-referencing to an average reference, eye artifacts were removed using independent component analysis (Infomax ICA) whose weights were calculated on the data band-pass filtered between 1 and 45 Hz (fourth-order Butterworth filter applied forward and backward). For SEP analysis, the data were segmented into epochs from –100 to 600 ms relative to stimulus onset, resulting in about 995 trials on average per participant. EEG pre-processing was

performed using EEGLAB (*Delorme and Makeig, 2004*), and custom-written scripts in MATLAB (The MathWorks Inc, Natick, MA).

## Single-trial extraction using CCA

Single-trial SEPs were extracted using canonical correlation analysis (CCA), as proposed by *Waterstraat et al., 2015*, and in the same way applied as described in *Stephani et al., 2020*, for a similar dataset.

CCA finds the spatial filters $w_x$ and $w_y$ for multi-channel signals $X$ and $Y$ by solving the following optimization problem for maximizing the correlation:

$$\max_{w_x, w_y} corr\left(w_x^T X, w_y^T Y\right),$$

where $X$ is a multi-channel signal constructed from concatenating all the epochs of a subject's recording, that is, $X = [x_1, x_2, \ldots, x_N]$ with $x_i \in \mathbb{R}^{channel \times time}$ being the multi-channel signal of a single trial and $N$ the total number of trials. Additionally, $Y = \underbrace{[\bar{x}, \ldots, \bar{x}]}_{N\ times}$ with $\bar{x} = \frac{1}{N}\sum_{i=1}^{N} x_i$ denoting the grand average of all trials. Since averaging cancels the background noise and recovers the shared morphology of the SEP of interest among all the trials, the CCA procedure resembles a template matching between the single-trial signals and the template time signature of the SEP of interest. The spatial filter $w_x$ provides us with a vector of weights for mixing the channels of each single trial (i.e., $x_{i,CCA} = w_x^T x_i$) and recovering their underlying SEP. Therefore, $w_x$ can be interpreted as the spatial signature of the SEP of interest across all single trials. The optimization problem of CCA can be solved using eigenvalue decomposition. Therefore, multiple CCA spatial components can be extracted for each subject, being the eigenvectors of the corresponding eigenvalue decomposition. Since we are mainly interested in the early portion of the SEP, the two signal matrices $X$ and $Y$ were constructed using shorter segments from 5 to 80 ms post-stimulus. The extracted CCA spatial filter was, however, applied to the whole-length epochs from –100 to 600 ms. The signal resulting from mixing the single trial's channels using the CCA spatial filter $w_x$, that is, $x_{i,CCA} = w_x^T x_i$, is called a CCA component of that trial.

The spatial activity pattern of each CCA component was computed by multiplying the spatial filters $w_x$ by the covariance matrix of $X$, as $cov(X)w_x$, in order to take the noise structure of the data into account (*Haufe et al., 2014*). The CCA components whose spatial patterns showed a pattern of a tangential dipole over the central sulcus (typical for the N20-P35 complex) were selected for further analyses and referred to as *tangential CCA components*. Such a tangential CCA component was present in all subjects among the first two CCA components with the maximum canonical correlation coefficients. Since CCA solutions are insensitive to the polarity of the signal, we standardized the resulting tangential CCA components by multiplying the spatial filter by a sign factor, in the way that the N20 potential always appeared as a negative peak in the SEP.

Furthermore, in a sub-sample of 13 subjects, a CCA component could be identified among the first four CCA components, which showed a peak at around 15 ms post-stimulus (presumably the P15 component of the SEP) and a spatial pattern that was characterized by a central, outspread activation (in the following referred to as *thalamic CCA component*). Also here, the CCA components were standardized so that the P15 always appeared as a positive peak.

In order to additionally evaluate the later time course of the SEP (i.e., the lower and later frequency content), the spatial filter of the tangential CCA component was applied to EEG data temporally filtered between 0.5 and 45 Hz (apart from this, pre-processed in the same way as described above).

## SEP peak amplitudes and pre-stimulus oscillatory activity

N20 peak amplitudes were defined as the minimum value in single-trial SEPs of the tangential CCA components ± 2 ms around the latency of the N20 in the within-subject average SEP. P15 amplitudes were measured from the thalamic CCA components as the average amplitude in a time window ±1 ms around the latency of the P15 in the within-subject average SEP. N140 amplitudes were measured from the low-frequency-filtered EEG (0.5–45 Hz), after application of the tangential CCA filter, as the average voltage in a time window between 140 and 160 ms after stimulus onset.

To estimate the average amplitude of pre-stimulus alpha band activity, the data were segmented from –500 to –5 ms relative to stimulus onset and band-pass filtered between 8 and 13 Hz, using

a fourth-order Butterworth filter (applied forward and backward). In order to avoid filter-related edge artifacts, the data segments were mirrored before filtering to both sides (symmetric padding). Segmenting the data before filtering prevented any leakage from post-stimulus signals to the pre-stimulus time window. In order to examine pre-stimulus alpha band activity of the same sources as of the SEP, the spatial filter of the tangential CCA component was also applied to the pre-stimulus alpha data. Subsequently, the amplitude envelope of the extracted alpha oscillations was computed by taking the absolute value of the analytic signal, using Hilbert transform of the real-valued signal. To derive one pre-stimulus alpha metric for every trial, amplitudes of the alpha envelope were averaged in the pre-stimulus time window of interest between –200 and –10 ms and log-transformed for subsequent statistical analyses in order to approximate a normal distribution.

### EEG source reconstruction

Sources of the EEG signal were reconstructed using lead field matrices based on individual brain anatomies and individually measured electrode positions. Structural T1-weighted MRI images (MPRAGE) were segmented using the Freesurfer software (http://surfer.nmr.mgh.harvard.edu/), and a three-shell boundary element model based on the segmented MRI was used to compute the lead field matrix with OpenMEEG (*Gramfort et al., 2010*; *Kybic et al., 2005*). A template brain anatomy (ICBM152; *Fonov et al., 2009*) was used for two subjects for whom no individual MRI scans were available. Additionally, standard electrode positions were used for one subject for whom the 3D digitization of the electrode positions was corrupted. The lead field matrices, constrained to sources perpendicular to the cortex surface, were inverted using the eLORETA method (*Pascual-Marqui, 2007*), and sources were reconstructed for the spatial patterns of the tangential CCA component of every subject, as well as for pre-stimulus alpha activity (on a single-trial level). For group-level analysis, we projected the individual source estimates onto the ICBM152 template anatomy using the spherical co-registration with the FSAverage template (*Fischl et al., 1999*) derived from Freesurfer. Subsequently, the source estimates were averaged across subjects. Brainstorm (*Tadel et al., 2011*) was used for building individual head models and visualizing the source space data. The MATLAB implementation of the eLORETA algorithm was derived from the MEG/EEG Toolbox of Hamburg (METH; https://www.uke.de/english/departments-institutes/institutes/neurophysiology-and-pathophysiology/research/research-groups/index.html).

### Processing of peripheral electrophysiological data (median nerve CNAP and thumb CMAP)

Analogously to the EEG data, stimulation artifacts were cut out and interpolated between –2 and 4 ms relative to stimulus onset using Piecewise Cubic Hermite Interpolating Polynomials. To achieve a sufficient SNR of the short latency CNAP peak of only a few milliseconds duration on single-trial level, the data were high-pass filtered at 70 Hz (fourth-order Butterworth filter applied forward and backward). For the CMAP, no further filtering was necessary given the naturally high SNR of muscle potentials (mV range). Here, only a baseline correction was performed from –20 to –5 ms to account for slow potential shifts. For the CNAP, single-trial peak amplitudes were extracted as the maximum amplitude ±1 ms around the participant-specific latency of the CNAP peak that was found between 5 and 9 ms in the within-participant averages. The CMAP was evaluated regarding its peak-to-peak amplitude, which was defined as the difference between the minimum and maximum amplitude measured ±1 ms around the participant-specific latencies of the negative and positive peaks of the biphasic CMAP response (which were found between 5 and 11 ms as well as 10 and 20 ms in the within-participant averages, respectively).

### Signal Detection Theory

In order to separate the discrimination ability of the two stimulus intensities from a general response bias, we applied Signal Detection Theory (SDT; *Green and Swets, 1966*; *Kingdom and Prins, 2016*). The ability to discriminate the two stimulus intensities was quantified using *sensitivity d'*, as calculated in the following way:

$$d' = \phi^{-1}\left(p\left(\text{"strong"}|strong\right)\right) - \phi^{-1}\left(p\left(\text{"strong"}|weak\right)\right),$$

where $\phi^{-1}$ corresponds to the inverse of the cumulative normal distribution, $p(\text{"strong"}|strong)$ to the probability of strong stimuli being rated as strong stimuli, and $p(\text{"strong"}|weak)$ to the probability of weak stimuli being rated as strong stimuli. Response probabilities were calculated as the number of responses divided by the number of stimuli of the respective categories. The response bias, *criterion c*, was calculated as follows:

$$c = -0.5 * \left( \phi^{-1} \left( p\left(\text{"strong"}|strong\right)\right) + \phi^{-1}\left(p\left(\text{"strong"}|weak\right)\right) \right).$$

According to SDT, *sensitivity d'* here represents the distance between the distributions of the internal responses of the two stimuli, and thus reflects the discriminability between strong and weak stimulus intensity. *Criterion c* reflects the internal threshold above which a stimulus is rated as strong stimulus and below which a stimulus is rated as weak stimulus, thus representing a general response bias. With respect to our data, a higher *criterion c* therefore indicates a general tendency to report lower stimulus intensities.

## Statistical analyses

To confirm that task accuracy was above chance level, we ran non-parametric permutation tests (*Crowley, 1992*). Within each participant, we derived a null distribution of chance-level performance by randomly remapping the behavioral responses with the presented stimuli 100,000 times (*Combrisson and Jerbi, 2015*). The p-value of the empirical task accuracy was calculated as the proportion of higher accuracy values in the surrogate data. Bonferroni correction was applied to account for the multiple tests across participants.

The effects of the EEG measures *pre-stimulus alpha amplitude*, *N20 peaik amplitude*, *P15 mean amplitude*, and *N140 mean amplitude* on the SDT measures *sensitivity d'* and *criterion c* were examined using a binning approach: First, trials were sorted according to the amplitudes of the EEG measures. Next, the SDT measures corresponding to the first and fifth quintile of the sorted trials were compared using paired-sample *t*-tests. To quantify effect sizes, *Cohen's d* was calculated as the mean difference between the dependent samples divided by the standard deviation of differences between the dependent samples.

The relationship between pre-stimulus alpha activity and the N20 component was tested using a random-slope linear-mixed effects model with *pre-stimulus alpha amplitude* as predictor of *N20 peak amplitude*, and *subject* as random factor:

*N20 peak amplitude* $\sim 1 + pre\text{-}stimulus\ alpha + (1 + pre\text{-}stimulus\ alpha\ |\ subject)$.

The relationship between thalamus-related activity and intensity perception was tested using a random-intercept linear-mixed-effects model with *P15 amplitude* and *presented stimulus intensity* as predictors of *perceived stimulus intensity*, as well as *subject* as random factor:

*Perceived stimulus intensity* $\sim 1 + P15\ amplitude + presented\ stimulus\ intensity + (1\ |\ subject)$.

Here, a logit link function was used to account for the dichotomous scale of *perceived stimulus intensity* (note that we refrained from estimating a random slope for P15 amplitude here given the small sample size of available data for thalamic activity). Furthermore, we tested the association between *presented stimulus intensity* and *P15 amplitude* as well as *pre-stimulus alpha amplitude*:

*P15 amplitude* $\sim 1 + pre\text{-}stimulus\ alpha + presented\ stimulus\ intensity + (1\ |\ subject)$.

Analogously, we analyzed the effect of N140 amplitude on perceived stimulus intensity, however now including random slopes for N140 amplitude and presented stimulus intensity:

*Perceived stimulus intensity* $\sim 1 + N140\ amplitude + presented\ stimulus\ intensity$

$+ (1 + N140\ amplitude + presented\ stimulus\ intensity\ |\ subject)$.

The dependence of the N140 on presented stimulus intensity and pre-stimulus alpha activity was examined using the following model:

*N140 amplitude* $\sim 1 + pre\text{-}stimulus\ alpha + presented\ stimulus\ intensity +$

$(1 + pre\text{-}stimulus\ alpha + presented\ stimulus\ intensity\ |\ subject)$.

**Table 2.** Relationships included in the hypothesized structural equation model ('SEM 1').
Level 1 equations reflect the within-participant effects between variables of interest. On level 2, only intercepts and variances of each variable were modeled; apart from *stimulus intensity* which only varied within participants by experimental design.

| Level 1 (within participants): |
| --- |
| N20 amplitude ~ 1 + stimulus intensity + pre-stimulus alpha |
| CNAP ~ 1 + stimulus intensity |
| CMAP ~ 1 + stimulus intensity |
| Perceived intensity ~ 1 + stimulus intensity + N20 amplitude + pre-stimulus alpha + CMAP |

| Level 2 (between participants): |
| --- |
| N20 amplitude ~~ N20 amplitude |
| CNAP ~~ CNAP |
| CMAP ~~ CMAP |
| Perceived intensity ~~ perceived intensity |
| Pre-stimulus alpha ~~ pre-stimulus alpha |

Furthermore, we conducted a post hoc power analysis to evaluate the probability of finding an effect of P15 amplitude on perceived stimulus intensity if it was existent. For this, we used Monte Carlo simulations with 1000 permutations based on the empirical dataset (*Green et al., 2016*), assuming an effect size of $\beta = 0.05$, which is in the range of the observed effect of N20 amplitude on perceived stimulus intensity.

In addition, the interrelation of pre-stimulus alpha activity, the N20 component of the SEP, peripheral nerve activity as measured by CNAP and CMAP, the presented stimulus intensity, as well as the perceived stimulus intensity were examined using confirmatory path analysis based on multi-level structural equation modeling as implemented in the general latent variable framework of *Mplus* (*Muthén and Muthén, 2017*). *Pre-stimulus alpha amplitude* and *presented stimulus intensity* were included as exogenous variables, *N20 peak amplitude*, *CNAP amplitude*, *CMAP amplitude*, and *perceived stimulus intensity* as endogenous variables. The relationships contained in the hypothesized model ('SEM 1') are summarized in *Table 2*. Trials with no behavioral response were excluded from the analysis. In total, 31,347 single trials were included in the SEM, with 979.6 trials on average per participant. Model parameters were estimated using the MLR estimator provided by *Mplus*, a maximum-likelihood estimator robust to violations of the assumption of normally distributed data. A logit link function was used to account for the dichotomous scale of *perceived stimulus intensity*. The fit of the hypothesized model was examined comparing it to alternative models constructed by step-wise including or excluding relevant effect paths (*Table 1*). Model comparisons were evaluated using $\chi^2$ difference tests (based on the log-likelihood; *Muthén, 2004*), the AIC, and the BIC. (Note that no other fit indices, such as CFI, RMSEA, or SRMR, are available for our type of model with a multi-level structure and a dichotomous outcome variable.)

Moreover, the association between pre-stimulus alpha activity, the N20 potential and perceived intensity were examined in source space, independently from the SEP-derived spatial CCA filter. For that, the sources of pre-stimulus alpha activity were reconstructed as described under *EEG source reconstruction*, and subjected to the following linear-mixed-effects models (in close correspondence to the sub-equations of the SEM approach):

$$N20\ peak\ amplitude \sim 1 + presented\ stimulus\ intensity + pre\text{-}stimulus\ alpha_{vertex\ i} + (1\mid subject),$$

and

$$Perceived\ stimulus\ intensity \sim 1 + presented\ stimulus\ intensity + pre\text{-}stimulus\ alpha_{vertex\ i} + (1\mid subject),$$

where *vertex i* refers to one of 5003 modelled cortical sources. We used FDR-correction (p < 0.01) to account for the multiple comparisons. Analogously, we repeated the SDT analysis for every single

source as well as for the averaged activity of a region of interest (ROI) that included 19 sources covering the hand region of the right primary somatosensory cortex (*Figure 5—figure supplement 1*).

For all analyses (apart from the FDR correction for multiple comparisons), the statistical significance level was set to p = 0.05 (two-sided). Correspondingly, two-sided confidence intervals were calculated with a confidence level of 0.95 (CI$_{95\%}$). The permutation-based analyses and *t*-tests were performed in MATLAB (version 2019b, The MathWorks Inc, Natick, MA). For both the linear-mixed-effects model and the structural equation models, all continuous variables (i.e., pre-stimulus alpha, N20 amplitude, CNAP, and CMAP) were z-transformed prior to statistics. The linear-mixed-effects models were calculated in R (version 3.5.3, *R Development Core Team, 2018*) with the *lmer* function of the *lme4* package (version 1.1–23, *Bates et al., 2015*), estimating the fixed-effect coefficients based on maximum likelihood. To derive a p-value for the fixed-effect coefficients, the denominator degrees of freedom were adjusted using Satterthwaite's method (*Satterthwaite, 1946*) as implemented in the R package *lmerTest* (version 3.1–2, *Kuznetsova et al., 2017*). Structural equation modeling was performed in *Mplus* (version 8.6, Base Program and Combination Add-On; *Muthén and Muthén, 2017*) using the *MplusAutomation* package in R for scripting (*Hallquist and Wiley, 2018*). Post hoc statistical power analyses were performed using the R package *simr* (*Green et al., 2016*).

## Acknowledgements

VVN was supported in part by the Basic Research Program at the National Research University Higher School of Economics. We thank Sylvia Stasch for participant recruitment and help with data collection.

## Additional information

### Funding

| Funder | Grant reference number | Author |
| --- | --- | --- |
| National Research University Higher School of Economics | | Vadim V Nikulin |

The funders had no role in study design, data collection and interpretation, or the decision to submit the work for publication.

### Author contributions

Tilman Stephani, Conceptualization, Data curation, Formal analysis, Investigation, Methodology, Software, Visualization, Writing – original draft, Writing – review and editing; Alice Hodapp, Mina Jamshidi Idaji, Formal analysis, Writing – review and editing; Arno Villringer, Funding acquisition, Resources, Supervision, Writing – review and editing; Vadim V Nikulin, Conceptualization, Investigation, Project administration, Resources, Supervision, Writing – review and editing

### Author ORCIDs

Tilman Stephani http://orcid.org/0000-0003-3323-3874
Alice Hodapp http://orcid.org/0000-0002-7886-0049
Mina Jamshidi Idaji http://orcid.org/0000-0003-1593-3201
Arno Villringer http://orcid.org/0000-0003-2604-2404
Vadim V Nikulin http://orcid.org/0000-0001-6082-3859

### Ethics

Human subjects: All participants gave informed consent and consent to publish. The study was approved by the local ethics committee (Ethical Committee at the Medical Faculty of Leipzig University, 04006 Leipzig, Germany).

### Decision letter and Author response

Decision letter https://doi.org/10.7554/eLife.67838.sa1
Author response https://doi.org/10.7554/eLife.67838.sa2

## Additional files

### Supplementary files
• Transparent reporting form

### Data availability

The data used for generating the figures are openly accessible at: https://osf.io/v9xa6/. The raw EEG data cannot be made available in a public repository due to the privacy policies for human biometric data according to the European General Data Protection Regulation (GDPR). Further preprocessed data can be obtained from the corresponding author (TS; stephani@cbs.mpg.de) upon reasonable request and as far as the applicable data privacy regulations allow it.

The following dataset was generated:

| Author(s) | Year | Dataset title | Dataset URL | Database and Identifier |
|---|---|---|---|---|
| Stephani T, Hodapp A, Jamshidi Idaji M, Villringer A, Nikulin VV | 2021 | Neural excitability and sensory input determine intensity perception with opposing directions in initial cortical responses | https://osf.io/v9xa6/ | Open Science Framework, v9xa6 |

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
