## [Decision Letter]

**Acceptance summary:**

Stephani et al., address the question of how ongoing fluctuations in neuronal excitability, as well as stimulus strength, impact the perception of above-threshold tactile stimuli and the subsequent stimulus-evoked brain activity. The study builds up on very high quality data as well as analysis approaches and, alongside a decent sample size and a host of additional peripheral measures and a simulation study, allows the authors to challenge common interpretations of brain potential magnitudes in stimulus-intensity encoding.

**Decision letter after peer review:**

Thank you for submitting your article "Neural excitability and sensory input determine intensity perception with opposing directions in initial cortical responses" for consideration by *eLife*. Your article has been reviewed by 2 peer reviewers, and the evaluation has been overseen by a Reviewing Editor and Floris de Lange as the Senior Editor. The following individuals involved in review of your submission have agreed to reveal their identity: Richard Gao (Reviewer #1); Nathan Weisz (Reviewer #2).

Essential revisions:

1) The main weaknesses of the manuscript become most apparent with respect to the stated impact that "The widespread belief that a larger brain response corresponds to a stronger percept of a stimulus may need to be revisited.". We are not sure that there are many cognitive neuroscientists who would subscribe to such a simplistic relationship between evoked responses and perception and that temporal differentiation (early vs late responses) and the biasing influence of prestimulus activity patterns are becoming increasingly recognized. So rather than actually changing a dominant paradigm, this work is an (excellent) contribution to a paradigm shift that is already taking place. Unneccesary claims of controiversiality and novelty thus should be toned down. See many specific hints to this in the two individual reviews below.

2) A main technical concern lies in the choice of decomposition filter for SEP and α oscillations, and the conclusions the authors draw from that. Specifically, a CCA spatial filter is optimized here for the N20 component, which is then identically applied to isolate for α sources, with the logic being that this procedure extracts the α oscillation from the same sources (e.g., L359). If our understanding of the authors' intent is correct, then the majority of us does not agree with the logic that using the same filter will isolate for α as well. The prestimulus α oscillation can have arbitrary source configurations that are different from the SEP sources, which may hypothetically have a different association with the behavioral responses when it's optimally isolated. In other words, just because one uses the same spatial filter, it does not imply that one is isolating α from the *same* source as the SEP, but rather simply projecting down to the same subspace – looking at a shadow on the same wall, if you will.

To show that they are from the same sources, α should be isolated independently of the SEP (using CCA, ICA, or other methods), and compared against the SEP topology. If the topology is similar, then it would strengthen the authors' current claims, but ideally the same analyses (e.g., using the 1st and 5th quintile of α amplitude to partition the responses) is repeated using α derived from this procedure. Also, have the authors considered using individualized α filters given that α frequency vary across individuals? Why or why not?

3) It should be considered that with regards to the analysis approach using CCA, the claims are mainly restricted to BA3b: The authors should refrain from overinterpreting the results in a very generalized manner. The authors do include some "thalamus" and "late" evoked response patterns as well, however that presentation of the results is somewhat changed now as compared to the N20 (e.g. using LMEs rather than comparison of extremes; not using SEMs). The readablity of results and especially the comparison of effects would profit from a more coherent approach.

4) Concerns arose whether the relationship between large α power and more negative N20s could be driven by more trivial factors rather than the model explanations the authors develop in the discussion. Put in concrete terms, the question is whether phase locking of large α power along with >30 Hz high pass filtering could produce a similar finding as shown e.g. in Figure 2c. This is an important issue, as prestimulus α influences the N20 amplitudes as well as the perceptual reports. See also our point #1 above.

5) At multiple points, the authors comment that the covariation of N20 and α amplitude in the same direction is counterintuitive (e.g., L123-125). It is being explained later in the manuscript that lower α amplitude and higher SEP amplitude are associated with excitability, and hence should have the opposite directions. This should be explicitly stated earlier in the introduction, as well as the expected relationship between α amplitude and behavior.

*Reviewer #1 (Recommendations for the authors):*

General comments for improving clarity and other discussion points

– Figure 1 is a really nice schematic of the study. Personally, it would have been extremely helpful to have an additional panel that shows the quintile analysis, and perhaps a visual representation of the quantities extracted (i.e., sensitivity and criterion). A slightly adapted version of a generic SDT schematic would be sufficient for the naive reader such as myself (as I had to Google around a bit), but could be unnecessary given the target audience of the paper

– What is the rationale for choosing the top and bottom 20% of α / SEP amplitude for partitioning? Is this just an arbitrary choice? If so, is the result robust to using, say, quartiles?

– α amplitude was higher in general when response was "weak", but SEP amplitude was larger for all stronger stimuli, and only larger when response was "weak" conditioned on the stimulus strength. In fact, it looks like the SEP effect is really driven by a difference within the strong stimuli (Figure 2d). I may have missed this, but a comment on this and why it might be would be great.

– I'm not sure if this was by design to emphasize the counterintuitiveness of the findings, but at multiple points in the manuscript, the description of the α and SEP effects are reversed with double negatives. For example, L100-102 says α amplitude was higher when participants rated the stim to be "weaker", while L140-141 says SEP was smaller when participants rated the stim to be "stronger". If I'm not mistaken, these mean the same thing, especially considering that N20 negativity is also intuitively an amplitude, in which case, both amplitude measures correlate with a bias towards "weaker" ratings. Same with the arrangement of figure 2b and 2e, where it would be natural to put higher amplitude (i.e., more negative N20) on the right. Unless this was done very intentionally, I think it would improve the clarity of the manuscript to make those descriptions and figures the same.

– Which is SEM1? Is that represented in Figure 4? I don't think this is labeled anywhere, and in the table SEM1 is just "original model"

– I might have misunderstood something here, but it was very surprising that the thalamic component does not vary as a function of stimulus intensity? Is there an explanation for this?

– Just to reiterate the earlier point on α, L312-317 is a really nice explanation of the model the authors have in mind, though I had a totally different one reading through the paper, that basically α amp and SEP amp have a shared origin, and hence both reflect synchronous neuronal activity, potentially of the same population. My reading of the author's interpretation is that α is a readout of excitability, and the lower the α amp the higher the excitability, which leads to more depolarized neurons and hence smaller EPSCs in a different (population). If this is correct, my suggestion would be to hint at this earlier on, so people don't go down the same rabbit hole as I did (though I am unfamiliar with this literature)

– Related to the above, I had to really wrap my head around the fact that the N20 is both a function of the true stimulus intensity (i.e. post-stim readout), as well as a marker for prestim (or ongoing) neural excitability (i.e., pre-stim readout). I know this is the whole point of the paper, but a sentence explicitly stating that very early on might help the naive reader to orient themselves to the subsequent findings.

– L328-330: nice emphasis on the significance, would do the same earlier as well if not already.

– I think a brief discussion on how the brain then uses the information represented by the SEP downstream to "perform the perceptual rating" would be great, i.e., does it account for the ongoing fluctuation in excitability?

– L351-355: CNAP (sensory readout) variability did not influence perceptual decision, but CMAP (muscle output) variability does – both are within the first 10ms, so there must be another loop from the motor reaction / reflex readout to the brain, and when does the brain / subject make the decision? Does the motor percept also feed back into the N20?

– I was looking for an explanation as to why criterion might be affected here but not sensitivity, maybe a brief discussion speculating on this point would be appropriate?

Again, great paper – it was a joy to read!

*Reviewer #2 (Recommendations for the authors):*

Overall I found the manuscript very interesting and my recommendations pretty much concern the discussed weaknesses.

– The authors should reflect more clearly the impact of the study along the lines mentioned above. I pretty much have the impression that the results are not a "game changer" in the field but fits well with increasingly dominant views in the field.

– I would strongly advise to check the possibility whether the α-N20-link could be caused by high-pass-filtering of phase locked α responses. Along these lines a time frequency depiction of results in figure 1a would be helpful (i.e. showing low to high frequencies of non-high pass filtered data). Also since the link between N20 and perception appears "counterintuitive", it may be also useful to test the relationship within α bins (i.e. high-low N20 responses stratified for α power) and see whether the relationship still holds.

– Furthermore, it would improve readability if the authors used a streamlined statistical approach also for the "thalamic" and "late" evoked responses.

– The authors convincingly show that evoked peripheral responses are not linked to N20 amplitude. It would be interesting to know whether variability in ongoing peripheral activity is linked to ongoing α activity (i.e. in prestimulus period).

– At the moment the value of the physiological model developed in the discussion is not really clear (see previous comments). It also seems to be particularly geared to the N20 results and leaves open the question whether it also encompasses the "thalamus" and "late" results.

---

## [Author Response]

Essential revisions:1) The main weaknesses of the manuscript become most apparent with respect to the stated impact that "The widespread belief that a larger brain response corresponds to a stronger percept of a stimulus may need to be revisited.". We are not sure that there are many cognitive neuroscientists who would subscribe to such a simplistic relationship between evoked responses and perception and that temporal differentiation (early vs late responses) and the biasing influence of prestimulus activity patterns are becoming increasingly recognized. So rather than actually changing a dominant paradigm, this work is an (excellent) contribution to a paradigm shift that is already taking place. Unneccesary claims of controiversiality and novelty thus should be toned down. See many specific hints to this in the two individual reviews below.

Thank you for this feedback. We agree that the paradigm shift away from simplistic assumptions about the relationship between variability of neural responses and perception is already taking place and that this is already being appreciated by many scientists in the field. Also, we agree that the present study contributes more evidence to this emerging notion rather than changing the whole field. However, we do think that particularly the observation of opposite amplitude modulations of initial somatosensory evoked responses associated with presented stimulus intensity on the one hand and pre-stimulus excitability state on the other, provides a novel perspective for our understanding of how fundamental features of sensory stimuli are processed at initial cortical levels. Following your suggestions to tone down claims about the controversiality as well as to avoid over-generalization, we have therefore adjusted the impact statement of this manuscript to:

“Larger evoked responses during initial cortical processing may reflect states of lower excitability.”

Furthermore, we have adjusted similar statements throughout the manuscript accordingly:

“This challenges previous assumptions that the amplitude of brain potentials, especially at early processing stages, reflects the coding of the perceived stimulus intensity.” (page 19, lines 360 ff.)

And

“Further questioning previous assumptions of how the evaluation of stimulus intensity is reflected in brain potentials, cortical excitability and the presented stimulus intensity were associated with opposing effects on the early SEP.” (page 22, lines 449 ff.)

2) A main technical concern lies in the choice of decomposition filter for SEP and α oscillations, and the conclusions the authors draw from that. Specifically, a CCA spatial filter is optimized here for the N20 component, which is then identically applied to isolate for α sources, with the logic being that this procedure extracts the α oscillation from the same sources (e.g., L359). If our understanding of the authors' intent is correct, then the majority of us does not agree with the logic that using the same filter will isolate for α as well. The prestimulus α oscillation can have arbitrary source configurations that are different from the SEP sources, which may hypothetically have a different association with the behavioral responses when it's optimally isolated. In other words, just because one uses the same spatial filter, it does not imply that one is isolating α from the same source as the SEP, but rather simply projecting down to the same subspace – looking at a shadow on the same wall, if you will.To show that they are from the same sources, α should be isolated independently of the SEP (using CCA, ICA, or other methods), and compared against the SEP topology. If the topology is similar, then it would strengthen the authors' current claims, but ideally the same analyses (e.g., using the 1st and 5th quintile of α amplitude to partition the responses) is repeated using α derived from this procedure. Also, have the authors considered using individualized α filters given that α frequency vary across individuals? Why or why not?

Indeed, applying the same spatial filter to EEG signals with different spatial arrangements of the sources can lead to the extraction of neuronal activity which does not originate from the very same sources. We had chosen our approach, as it is well known that the generators of the early SEP components and the generators of the prominent somatosensory α rhythm co-reside at similar sites in the primary somatosensory cortex (e.g., Haegens et al., 2015). Therefore, we considered our approach appropriate to specifically focus on neural activity from the somatosensory region both in the frequency band of the SEP as well as of the α rhythm. Yet, we agree with the reviewer that it should be acknowledged that we may have missed or mixed-up effects of α activity from other sources by using this procedure (which might have led to different conclusions otherwise). In order to account for this, we repeated our analyses with an SEP-independent reconstruction of the oscillatory effects in source space (“whole brain analysis”). For this, we first reconstructed the sources of α activity using eLORETA and head models based on participant-specific MRI scans, and estimated the respective effects independently for all sources across the cortex using both linear-mixed effects models (LME) as well as a binning approach for the Signal Detection Theory (SDT) parameters *sensitivity d’* and *criterion c* (consistent with the previous analyses in our manuscript). In the LME analyses, both the effects of pre-stimulus α activity on N20 amplitudes as well as on perceived stimulus intensity were strongest in the right primary somatosensory cortex – in accordance with the sources of the originally extracted tangential CCA component of the SEP (see Figure 5). Also, using the binning approach to examine the relation or pre-stimulus α activity with SDT parameter *criterion c*, the effects were most pronounced around the right somatosensory regions (see Figure 5—figure supplement 1), yet these effects did not survive statistical correction for multiple comparisons (FDR-correction with *p*<.01). However, when performing the same binning analysis for our region of interest (ROI), the hand area in BA 3b of the right somatosensory cortex, a significant effect or pre-stimulus α on *criterion c* was indeed confirmed, *t*(31)=-2.951, *p*=.006, CI_95%_=[-.173, -.032]. Furthermore, in line with our previous CCA results, for *sensitivity d’*, neither the whole brain analysis nor the ROI analysis showed effects of pre-stimulus α amplitude, *t*(31)=0.633, *p*=.531, CI_95%_=[-.083,.157].

Taken together, the findings we report in our original manuscript for pre-stimulus α activity obtained with the spatial CCA filter can thus be replicated with a SEP-uninformed source reconstruction, both using LMEs for a “whole-brain analysis” as well as SDT analyses in a ROI-based approach. We therefore conclude that the relationships between pre-stimulus α activity, N20 potential of the SEP, and perceived stimulus intensity can indeed be attributed to neural activity from the same (or at least very similar) sources in the primary somatosensory cortex.

We have added the following sections to our manuscript where we describe these analyses in more detail:

Results (page 12, lines 206 ff.):

“Reconstruction of the observed effects in source space

In order to investigate whether the observed effects of pre-stimulus α activity on N20 amplitudes and the perceived stimulus intensity were specific only to the generator regions of the SEP, we repeated the SDT analysis as well as the linear-mixed-effects models for these relations in source space (i.e., separately for every source estimated based on individual head models; see Methods). As visible from Figure 5, the effects of pre-stimulus α amplitude on both N20 amplitude as well as perceived stimulus intensity were indeed most pronounced around the hand region of the right primary somatosensory cortex, the same region which we identified as source for the tangential CCA component used in the analyses above. The effects in source space using the SDT approach did not reach significance after the correction for multiple comparisons. Yet, a ROI analysis within the hand region of the right primary somatosensory cortex did confirm the observations from our previous analyses: There was an effect of pre-stimulus α amplitude on SDT parameter criterion c, t(31)=-2.951, p=.006, CI95%=[-.173, -.032], but no effect on sensitivity d’, t(31)=0.633, p=.531, CI95%=[-.083,.157]. (Please refer to Figure 5—figure supplement 1 for the distribution of the SDT effects across the whole cortex.) Taken together, the relationships between pre-stimulus α activity, N20 potential of the SEP, and perceived stimulus intensity appear to be attributable to neural activity from the same (or at least very similar) sources in the right primary somatosensory cortex.”

Discussion (page 20, lines 406 ff.):

“An additional, SEP-independent analysis of the effects of pre-stimulus α amplitude on N20 amplitude and perceived stimulus intensity confirmed these findings in source space.”

Methods:

“[…] sources were reconstructed for the spatial patterns of the tangential CCA component of every subject, as well as for pre-stimulus activity filtered in the α band (8 to 13 Hz; extracted on a single-trial level).” (page 30, lines 631 ff.)

and:

“Moreover, the association between pre-stimulus α activity, the N20 potential and perceived intensity were examined in source space, independently from the SEP-derived spatial CCA filter. For that, the sources of pre-stimulus α activity were reconstructed as described under *EEG source reconstruction*, and subjected to the following linear-mixed-effects models (in close correspondence to the sub-equation of the SEM approach):

*N20 peak amplitude* ~ 1 + *presented stimulus intensity* + *pre-stimulus alpha_vertex i_* + (1 | *subject*),

and

*Perceived stimulus intensity* ~ 1 + *presented stimulus intensity* + *pre-stimulus alpha_vertex i_* + (1 | *subject*)

where *vertex i* refers to one of 5003 modelled cortical sources. We used FDR-correction (*p*<.01) to account for the multiple comparisons. Analogously, we repeated the SDT analysis for every single source as well as for the averaged activity of a region of interest (ROI) that included 19 sources covering the hand region of the right primary somatosensory cortex (Figure 5—figure supplement 1).” (page 34, lines 735 ff.)

Addressing the question on filtering α activity in individualized frequency bands, we considered this option, too. However, the rather short length of our pre-stimulus window (-200 to -10 ms) constitutes a natural limit for the frequency resolution in the α range and slightly different filter ranges (adjusted with regards to the individual α peak frequency) are thus unlikely to lead to large differences in the estimation of pre-stimulus α amplitudes. Therefore, we refrained from using individualized frequency bands here and focused on the more generic approach using one common α band (8-13 Hz) for all participants, which should also facilitate direct comparisons with previous studies on pre-stimulus oscillatory effects.

3) It should be considered that with regards to the analysis approach using CCA, the claims are mainly restricted to BA3b: The authors should refrain from overinterpreting the results in a very generalized manner. The authors do include some "thalamus" and "late" evoked response patterns as well, however that presentation of the results is somewhat changed now as compared to the N20 (e.g. using LMEs rather than comparison of extremes; not using SEMs). The readablity of results and especially the comparison of effects would profit from a more coherent approach.

We agree that our findings indeed have the specific focus on the N20 component and thus on its generators in BA3b. We did not intend to suggest that the effects we observed for this initial cortical response can be readily generalized to other (later) ERP components, too. However, we do believe (and hypothesize) that similar mechanisms may be in place for corresponding initial cortical responses in other sensory modalities, too – yet it is clear that we cannot test this generalization with the current study. To avoid misunderstandings of these interpretations and their limitations and also relating to the last comment of reviewer #2, we have further specified these aspects in the Discussion; page 19, lines 373 ff.:

“In this context, it should be further emphasized that our proposed physiological model may be directly applicable to well isolated neural signals only (such as the N20 component of the SEP). The physiological interpretation of amplitudes of later EEG potentials, such as the N140, however, is not as straightforward as described above, since several distinct SEP components may interact (Auksztulewicz et al., 2012), and excitatory and inhibitory contributions cannot be readily distinguished.”

Regarding our analyses of the later SEP (i.e., N140 component) and thalamus-related activity (i.e., P15 component), we initially decided to use linear-mixed effects models as they are mathematically equivalent to the way the sub-equations of the structural equation model were constructed (Table 2 in the manuscript). Nevertheless, we have now additionally run binning analyses to make a direct comparison also with Signal Detection Theory (SDT) parameters possible: For the N140 component, there was a significant effect on *criterion c*, *t*(31)=-3.010, *p*=.005, but no effect on *sensitivity d’*, *t*(31)=0.246, *p*=.807. For the P15 component, no effects emerged either for *criterion c* or *sensitivity d’*, *t*(12)=1.201, *p*=.253, and *t*(12)=-0.201, *p*=.844, respectively. These findings correspond well to the previous LME analyses and may indeed further facilitate the comparison with the findings for the N20 potential and pre-stimulus α activity. Therefore, we have added these complimentary analyses to our manuscript in the following way:

Results:

“In addition, the SDT analysis based on binning of the P15 amplitudes into quintiles neither suggested a relation with *criterion c* nor with *sensitivity d’*, *t*(12)=1.201, *p*=.253, and *t*(12)=-0.201, *p*=.844, respectively.” (page 14, lines 241 ff.)

and

“These findings were in line with a separate SDT analysis: N140 amplitudes were associated with an effect on *criterion c*, *t*(31)=-3.010, *p*=.005, but no effect on *sensitivity d’* emerged, *t*(31)=0.246, *p*=.807.” (page 15, lines 263 ff.)

Discussion:

“Crucially, our data are at the same time consistent with previous studies on somatosensory processing at later stages, where larger EEG potentials are typically associated with a stronger percept of a given stimulus (e.g., Al et al., 2020; Schröder et al., 2021; Schubert et al., 2006), as both our SDT and LME analyses of the N140 component showed.” (page 19, lines 367 ff.)

and

“Yet, neither our SDT analyses nor the LME models of the thalamus-related P15 component supported this notion.” (page 21, lines 414 ff.)

Methods (page 32, lines 681 ff.):

“The effects of the EEG measures *pre-stimulus α amplitude*, *N20 peak amplitude*, *P15 mean amplitude*, and *N140 mean amplitude* on the SDT measures *sensitivity d’* and *criterion c* were examined using a binning approach: […]”

4) Concerns arose whether the relationship between large α power and more negative N20s could be driven by more trivial factors rather than the model explanations the authors develop in the discussion. Put in concrete terms, the question is whether phase locking of large α power along with >30 Hz high pass filtering could produce a similar finding as shown e.g. in Figure 2c. This is an important issue, as prestimulus α influences the N20 amplitudes as well as the perceptual reports. See also our point #1 above.

Indeed, potential phase-locking of α oscillations to stimulus onset and filter-related effects are important issues that could potentially offer an alternative explanation for the observed relationship between amplitudes of pre-stimulus α activity and the N20 potential of the SEP. Although such pre-stimulus α locking is rather unlikely in a paradigm with jittered stimulus onsets (in our case uniformly distributed between -50 ms and +50 ms; corresponding to a whole α cycle), we have run the following control analyses to fully exclude this possibility:

First, we analyzed whether pre-stimulus α phase values were distributed uniformly and whether these phase distributions differed between high and low α amplitudes as well as between high and low N20 amplitudes. The phase of pre-stimulus α activity was obtained from a Fast-Fourier transform in the pre-stimulus time window from -200 to -10 ms, applied to unfiltered, but otherwise identically pre-processed data as in the original manuscript (i.e., applying the spatial filter of the tangential CCA component). For the FFT, we used zero padding (extending the pre-stimulus data segments to 2048 data points each) in order to obtain an interpolated frequency resolution of around 3 Hz. The phase was extracted at the frequency 9.766 Hz (i.e., the closest available frequency to 10 Hz). As visible from Author response image 1, pre-stimulus α phases were distributed uniformly across all five quintiles of both α and N20 amplitudes. This observation was confirmed by the Rayleigh test (testing for deviations from a uniform distribution; Berens, 2009): Neither in the concatenated phase data of all participants, *z*=1.130, *p*=.323, nor in single-participant analyses within every α amplitude or N20 amplitude bin, we found evidence for a non-uniform distribution of α phase, all *p*>.367 (after Bonferroni correction for multiple testing). Thus, there was no phase-locking of pre-stimulus α activity that could serve as a trivial alternative explanation of the relationship between pre-stimulus α amplitude and N20 amplitude.

**Author response image 1. sa1fig1:** Phase distributions of pre-stimulus alpha band activity, separately displayed for quintiles of alpha amplitudes (panel a), and N20 amplitudes (panel b). Alpha phase estimates were based on Fast-Fourier transforms in the pre-stimulus time windows from -200 to -10 ms. For visualization, data were concatenated across all participants (N=32).

Second, in order to examine whether the combination of our temporal filters (30 to 200 Hz band-pass for the SEP, and 8 to 13 Hz band-pass for α activity) could have led to the present findings, we additionally re-ran our analysis pipeline with simulated data: We mixed exemplary SEP responses with constant amplitudes (unfiltered; derived from within-participant averages), with simulated α band activity with randomized amplitude fluctuations, and pink noise, reflecting neural background activity as is typical for the human EEG. The SEP onsets were chosen according to our original experimental paradigm with inter-stimulus intervals of 1513 ms and a jitter of ±50 ms. Next, we filtered these mixed signals between 30 and 200 Hz in order to extract the single-trial SEPs, and estimated the pre-stimulus α amplitudes between -200 and -10 ms in the same way as was done in the original manuscript (i.e., by filtering the mixed signal between 8 and 13 Hz). This procedure was repeated for 32 generated data streams, containing 1000 SEPs each (corresponding to our empirical dataset of 32 participants). As can be seen from Figure 2—figure supplement 3, the resulting average SEPs did neither show a visually detectable difference between the five α amplitude quintiles nor indicated a random-slope linear-mixed-effects model any relation between pre-stimulus α amplitude and N20 amplitude on a single-trial level, *β_fixed_*=-.0005, *t*(255.16)=-.094, *p*=.925. Therefore, our findings cannot be explained by filter artifacts or residual activity leaking from the α frequency band to the frequency band of the N20 potential.

Third, we re-analyzed our empirical EEG data in time-frequency space to obtain a more detailed view of the effects of pre-stimulus α activity on N20 amplitudes. For this, we decomposed our pre-processed but unfiltered data with wavelet transformation (complex Morlet wavelets) and calculated linear-mixed effects models on the relation between signal amplitudes in the time-frequency domain and single-trial N20 amplitudes as obtained from our original analyses. As shown in Figure 2—figure supplement 2, the time-frequency representations of the effects on N20 amplitudes indeed indicated a specific role of the α band, with its effects (i.e., already 200 ms before stimulus and in the upper α frequency range) separated from the time- and frequency range of the N20 potential of the SEP (i.e., from ~20 ms after stimulus onwards and above ~20 Hz). In addition, we ran the same analysis for the behavioral effect (i.e., perceived stimulus intensity). Also here, pre-stimulus effects were predominantly visible in the α band. Of note, there were also strong effects in the β band. These may be interesting to study further in future studies – in particular, whether they reflect independent physiological processes or rather harmonics of the α band. Furthermore, these time-frequency representations suggest that the studied pre-stimulus effects might have been even more pronounced if we had analyzed the data in pre-stimulus time windows from -300 to -10 ms. However, in order to avoid inflating effect sizes by post-hoc data digging (“p-hacking”), we prefer to keep the original, a priori chosen time window for the main analyses of the manuscript. Yet, these onsets of pre-stimulus effects at around -300 ms may be of interest for future work. Taken together, these time-frequency analyses further support the notion that the observed relation between pre-stimulus α activity and N20 amplitudes is not due to technical issues (such as filter leakage and phase-locking) but rather reflects genuine neurophysiological effects of α oscillations on SEPs.

We have added the time-frequency analysis (Figure 2 Supplement 3), as well as the SEP simulation analysis (Figure 2 Supplement 4) as figure supplements to Figure 2 in our revised manuscript (page 8) since we believe that these control analyses comprehensively show that the observed effects were (a) specific to the α band and (b) not due to any data processing-related artifacts.

5) At multiple points, the authors comment that the covariation of N20 and α amplitude in the same direction is counterintuitive (e.g., L123-125). It is being explained later in the manuscript that lower α amplitude and higher SEP amplitude are associated with excitability, and hence should have the opposite directions. This should be explicitly stated earlier in the introduction, as well as the expected relationship between α amplitude and behavior.

Thank you for pointing out this unclarity. We have now made this rationale more explicit already at an early point in the introduction (page 3, lines 26 ff.):

“According to the baseline sensory excitability model (BSEM; Samaha et al., 2020), higher α activity preceding a stimulus indicates a generally lower excitability level of the neural system, resulting in smaller stimulus-evoked responses, which are in turn associated with a lower detection rate of near-threshold stimuli but no changes in the discriminability of sensory stimuli (since neural noise and signal are assumed to be affected likewise).”

Reviewer #1 (Recommendations for the authors):General comments for improving clarity and other discussion points– Figure 1 is a really nice schematic of the study. Personally, it would have been extremely helpful to have an additional panel that shows the quintile analysis, and perhaps a visual representation of the quantities extracted (i.e., sensitivity and criterion). A slightly adapted version of a generic SDT schematic would be sufficient for the naive reader such as myself (as I had to Google around a bit), but could be unnecessary given the target audience of the paper.

Good idea! We have added a SDT schematic as Figure 2—figure supplement 1.

– What is the rationale for choosing the top and bottom 20% of α / SEP amplitude for partitioning? Is this just an arbitrary choice? If so, is the result robust to using, say, quartiles?

The choice of partitioning the data into quintiles was made a priori, based on previous studies in the field (e.g., Iemi et al., 2017). In addition, we repeated the analysis with quartiles which resulted in very similar observations: There were significant effects for *criterion c* for both the pre-stimulus α and N20 analysis, *t*(31)=-3.238, *p*=.003, and *t*(31)=2.434, *p*=.021, respectively (always comparing first vs. fourth bin, sorted in ascending order). However, no effects emerged for *sensitivity d’* either for α or N20 amplitude, *t*(31)=-.672, *p*=.507, or N20 amplitude, *t*(31)=.001, *p*>.999. Therefore, our findings appear robust also with other ways of partitioning the data.

– α amplitude was higher in general when response was "weak", but SEP amplitude was larger for all stronger stimuli, and only larger when response was "weak" conditioned on the stimulus strength. In fact, it looks like the SEP effect is really driven by a difference within the strong stimuli (Figure 2d). I may have missed this, but a comment on this and why it might be would be great.

Indeed, the SEP effect on behavior seemed to be driven by trials in which the strong stimulus intensity was presented (although the effect direction was the same also for weak stimuli). As this may be an important observation to be considered for the design of future studies, we have added this aspect with some tentative explanations (page 9, lines 148 ff.):

“Interestingly, the relationship between N20 amplitudes and perceptual outcome appeared to be driven mainly by differences within the strong stimulus category (Figure 2d). This may reflect the naturally higher signal-to-noise ratio of SEPs in response to stronger stimuli, or it could also point out that there was a “floor effect” for the modulation of SEPs in response to the weaker stimuli.”

– I'm not sure if this was by design to emphasize the counterintuitiveness of the findings, but at multiple points in the manuscript, the description of the α and SEP effects are reversed with double negatives. For example, L100-102 says α amplitude was higher when participants rated the stim to be "weaker", while L140-141 says SEP was smaller when participants rated the stim to be "stronger". If I'm not mistaken, these mean the same thing, especially considering that N20 negativity is also intuitively an amplitude, in which case, both amplitude measures correlate with a bias towards "weaker" ratings. Same with the arrangement of figure 2b and 2e, where it would be natural to put higher amplitude (i.e., more negative N20) on the right. Unless this was done very intentionally, I think it would improve the clarity of the manuscript to make those descriptions and figures the same.

Thank you for this feedback. We have unified the wording throughout the manuscript so that there are no “double negatives” anymore, which will hopefully make the meaning easier to grasp. The arrangement of Figure 2b and 2e, however, was indeed intentional since we wanted to emphasize the unexpected direction of the N20 effects, and also, to keep the order of data partitions (“bins”) the same across panels (i.e., in ascending order). We have made the following changes:

“Thus, c*riterion c* was lower for smaller than for larger N20 peak amplitudes (Figure 2e). This indicates that participants were more likely to rate a stimulus as “strong” rather than “weak” when the magnitude of the N20 potential was smaller (i.e., less negative), after taking into account the stimulus´ actual intensity, as is it also becomes evident from the SEPs sorted by the behavioral response categories (Figure 2d).” (page 9, lines 143 ff.)

and:

“When controlling for stimulus intensity, both higher pre-stimulus α amplitudes and larger (i.e., more negative) N20 amplitudes were associated with a lower perceived intensity (equivalent to a response bias as reflected in *criterion c*), as well as higher pre-stimulus α amplitudes co-occurred with larger (i.e., more negative) N20 amplitudes.” (page 11, lines 184 ff.)

as well as:

“Moreover, the notion of smaller (i.e., less negative) N20 amplitudes reflecting a state of higher excitability is corroborated by the behavioral data: When controlling for stimulus intensity, we found smaller N20 amplitudes to be associated with higher perceived stimulus intensity.” (page 18, lines 349 ff.)

– Which is SEM1? Is that represented in Figure 4? I don't think this is labeled anywhere, and in the table SEM1 is just "original model".

Yes, “SEM1” is the “original model”. Thank you for spotting this inconsistency. We have clarified it throughout the whole manuscript (see Figure 4 (page 10), Table 1 (page 12), and Table 2 (page 34)).

– I might have misunderstood something here, but it was very surprising that the thalamic component does not vary as a function of stimulus intensity? Is there an explanation for this?

Good point – indeed, Figure 5a does not suggest any effect of stimulus intensity on P15 amplitude, which is puzzling. We confirmed this by a random-intercept linear-mixed-effects model (i.e., P15 amplitude as dependent variable and stimulus intensity as predictor), which could not find an effect of stimulus intensity on P15 amplitude, *β*_stim_int_=.009, *t*(12721.82)=.531, *p*=.596 (CI_95%_ of *β*_stim_int_: [-.025,.043]). (As a side note: Here, it did also not make a noteworthy difference whether pre-stimulus α amplitude was included as a co-variate or not.) The lack of this (expected) effect of stimulus intensity on the P15 may be associated with the nucleus-like structure of the thalamus where the electrical signals from the neurons are not summed up as effectively as in the laminar structures of the cortex, resulting in a lower signal-to-noise ratio in the EEG. In addition, the deep location of the thalamus may have led to a considerable attenuation of the corresponding signals thus further decreasing the SNR. Furthermore, the difference between the weak and strong stimulus intensities was very small (adjusted according to the *least-noticeable difference*), making it even more challenging to detect difference effects in the EEG. Nevertheless, the lack of this intensity effect poses an important limitation for the interpretation of the thalamic control analysis, which we have added to the manuscript in the following way:

Results (page 14, lines 246 ff.):

“However, P15 amplitudes did also not differ between different presented stimulus intensities, as tested with a random-intercept linear-mixed-effects model, *β*_stim_int_=.009, *t*(12721.82)=.531, *p*=.596 (CI_95%_ of *β*_stim_int_: [-.025,.043]), which may indicate that this EEG-based measure of thalamic activity is generally not very sensitive to differences between experimental conditions. For completeness, we also tested for the effect of pre-stimulus α amplitude on P15 amplitude, which was also not significant, *β*_prestim_=-.007, *t*(12210.50)=-.635, *p*=.526 (CI_95%_ of *β*_prestim_: [-.028,.014]).”

Discussion (page 21, lines 415 ff.):

“Although we estimated an acceptable statistical power of these analyses, it should be noted that we could also not observe an effect of presented stimulus intensity on P15 amplitudes. Therefore, these results should be interpreted with some caution as this measure may lack the required signal-to-noise ratio to detect rather subtle experimental effects of interest such as of the intensity difference of two very similar stimuli. Nevertheless – and also taking into account our analyses in source space – we conclude that the findings of the present study are most consistent with the idea that the modulation of perceived intensity had its origins at the cortical level.”

Methods (page 32, lines 699 ff.):

“Furthermore, we tested the association between *presented stimulus intensity* and *P15 amplitude* as well as *pre-stimulus α amplitude*:

*P15 amplitude* ~ 1 + *presented stimulus intensity* + *pre-stimulus α* + (1 | *subject*).”

– Just to reiterate the earlier point on α, L312-317 is a really nice explanation of the model the authors have in mind, though I had a totally different one reading through the paper, that basically α amp and SEP amp have a shared origin, and hence both reflect synchronous neuronal activity, potentially of the same population. My reading of the author's interpretation is that α is a readout of excitability, and the lower the α amp the higher the excitability, which leads to more depolarized neurons and hence smaller EPSCs in a different (population). If this is correct, my suggestion would be to hint at this earlier on, so people don't go down the same rabbit hole as I did (though I am unfamiliar with this literature).

Thank you for this suggestion. We have now added a more detailed rationale behind this idea already at an early point in the introduction (in line with our response to *Essential Revision #5*; page 3, lines 26 ff.):

“According to the baseline sensory excitability model (BSEM; Samaha et al., 2020), higher α activity preceding a stimulus indicates a generally lower excitability level of the neural system, resulting in smaller stimulus-evoked responses, which are in turn associated with a lower detection rate of near-threshold stimuli but no changes in the discriminability of sensory stimuli (since neural noise and signal are assumed to be affected likewise).”

– Related to the above, I had to really wrap my head around the fact that the N20 is both a function of the true stimulus intensity (i.e. post-stim readout), as well as a marker for prestim (or ongoing) neural excitability (i.e., pre-stim readout). I know this is the whole point of the paper, but a sentence explicitly stating that very early on might help the naive reader to orient themselves to the subsequent findings.

Yes, this duality of the meaning of the N20 amplitude is exactly what we wanted to convey. We have made this idea more explicit already in the introduction in the following way:

“According to the baseline sensory excitability model (BSEM; Samaha et al., 2020), higher α activity preceding a stimulus indicates a generally lower excitability level of the neural system, resulting in smaller stimulus-evoked responses, which are in turn associated with a lower detection rate of near-threshold stimuli but no changes in the discriminability of sensory stimuli (since neural noise and signal are assumed to be affected likewise).” (page 3, lines 26 ff.)

as well as:

“The N20 component, a negative deflection after around 20 ms at centro-parietal electrode sites in response to median nerve stimulation, reflects excitatory post-synaptic potentials (EPSPs) of the first thalamo-cortical volley (Wikström et al., 1996; Nicholson Peterson et al., 1995; Bruyns-Haylett et al., 2017) which are generated in the anterior wall of the postcentral gyrus, Brodmann area 3b (Allison et al., 1991). Thus, the N20 directly reflects the intensity of a given stimulus. However, when keeping the sensory input constant, the amplitude of this early part of the SEP only depends on the excitability of the involved, well-defined neuronal population in the primary somatosensory cortex, and therefore represents an excellent instantaneous probe thereof.” (page 4, lines 46 ff.)

– L328-330: nice emphasis on the significance, would do the same earlier as well if not already.

Thank you for the suggestion. In line with *Essential Revision #1*, we have now toned down “unnecessary claims of controversiality” throughout the whole manuscript and would therefore be reluctant to put the mentioned statement at a more prominent place in the manuscript without a more differentiated discussion (which follows on page 19, lines 360 ff.):

“This challenges previous assumptions that the amplitude of brain potentials, especially at early processing stages, reflects the coding of the perceived stimulus intensity. Rather, our findings call for a more differentiated view. Although the amplitude of early event-related potentials may indeed reflect the size of the input (e.g., a stronger or weaker somatosensory stimulus), the neural evaluation of this input (i.e., the perceived intensity), however, further depends on internal neural states, such as neural excitability, which may even reverse the amplitude effects of the input already at the earliest cortical processing stages.”

– I think a brief discussion on how the brain then uses the information represented by the SEP downstream to "perform the perceptual rating" would be great, i.e., does it account for the ongoing fluctuation in excitability?

Exciting question! From the present study, we probably cannot tell in detail how the information at early stages is integrated in later, more complex neural processing (or whether there are even compensating mechanisms for variability of initial cortical activation). However, changes in early sensory processing should in principle provide the ground for these later processes: Regardless of how neural processing occurs at later stages, the only possibility for the neural system to discriminate between two stimuli is to use the initial sensory responses. We have added this discussion together with a slightly broader perspective on the integration of “noise” in neural systems in the following way (page 21, lines 432 ff.):

“In addition, it is unclear at this point how exactly the observed fluctuations of initial cortical responses are integrated in later, downstream neural processes. In principle, changes in early sensory processing should provide the ground for later neural activity involved in the perceptual decision making, and finally shape the behavioral outcome (as observed in the current study). However, with our data, we cannot unambiguously tell whether the modulation of α oscillations – associated with excitability changes at the earliest cortical level – may in turn reflect a top-down regulated signal, which could in principle enable the neural system to account for ongoing fluctuations of excitability and even benefit from a certain degree of variability. Although more and more evidence indeed suggest an adaptive, functional role of neural “noise” (e.g., Findling and Wyart, 2021), further studies are needed to better understand how this concept may pertain also to such fundamental neural properties as the initial cortical excitability to external sensory stimuli.”

– L351-355: CNAP (sensory readout) variability did not influence perceptual decision, but CMAP (muscle output) variability does – both are within the first 10ms, so there must be another loop from the motor reaction / reflex readout to the brain, and when does the brain / subject make the decision? Does the motor percept also feed back into the N20?

The thumb twitch in response to supra-threshold median nerve stimulation results from the excitation of motor axons at the site of stimulation (in our study the left wrist), thus activating muscle fibers via efferent neurons. Therefore, the CMAP is not necessarily related to the CNAP, which reflects the excitation of afferent neurons in response to the median nerve stimulation. Indeed, the afferent (i.e., proprioceptive) information of the thumb twitch itself should also travel along the median nerve – but this should happen markedly after the initial CNAP (and presumably in a less synchronized manner leading to a much lower measurable signal). This CMAP-related afferent signal then most likely influenced the perceived stimulus intensity as processed in the brain – however, in turn only at a later stage than the initial N20 response. Therefore, there is no possibility that the motor percept feeds back into the N20 signal. This also becomes evident from our SEM analyses where we did not observe a relation between N20 and CMAP amplitudes.

– I was looking for an explanation as to why criterion might be affected here but not sensitivity, maybe a brief discussion speculating on this point would be appropriate?

The idea of the baseline sensory excitability model (BSEM; Samaha et al., 2020) is that a generally lower excitability level already at early sensory processing stages results in a lower probability of reporting the presence of a stimulus no matter whether the stimulus was actually shown or not (i.e., decreasing both the numbers of hits and false alarms). This would correspond to a change in SDT parameter *criterion c*. The discriminability, that is SDT parameter *sensitivity d’*, however, should stay constant since the perception of noise and signal are changed by the excitability level in the same way. Thank you for pointing out this missing link which we have now added to the introduction (page 3, lines 26 ff.):

“According to the baseline sensory excitability model (BSEM; Samaha et al., 2020), higher α activity preceding a stimulus indicates a generally lower excitability level of the neural system, resulting in smaller stimulus-evoked responses, which are in turn associated with a lower detection rate of near-threshold stimuli but no changes in the discriminability of sensory stimuli (since neural noise and signal are assumed to be affected likewise).”

Again, great paper – it was a joy to read!Reviewer #2 (Recommendations for the authors):Overall I found the manuscript very interesting and my recommendations pretty much concern the discussed weaknesses.– The authors should reflect more clearly the impact of the study along the lines mentioned above. I pretty much have the impression that the results are not a "game changer" in the field but fits well with increasingly dominant views in the field.

Yes, we fully agree. We have tried to incorporate this in our changes in relation to *Essential Revision #1*, adjusting the impact statement as well as similar statements throughout the manuscript:

Impact statement:

“Larger evoked responses during initial cortical processing may reflect states of lower excitability.”

Statements in the manuscript text:

“This challenges previous assumptions that the amplitude of brain potentials, especially at early processing stages, reflects the coding of the perceived stimulus intensity.” (page 19, lines 360 ff.)

and

“Further questioning previous assumptions of how the evaluation of stimulus intensity is reflected in brain potentials, cortical excitability and the presented stimulus intensity were associated with opposing effects on the early SEP.” (page 22, lines 449 ff.)

– I would strongly advise to check the possibility whether the α-N20-link could be caused by high-pass-filtering of phase locked α responses. Along these lines a time frequency depiction of results in figure 1a would be helpful (i.e. showing low to high frequencies of non-high pass filtered data). Also since the link between N20 and perception appears "counterintuitive", it may be also useful to test the relationship within α bins (i.e. high-low N20 responses stratified for α power) and see whether the relationship still holds.

Thank you for these suggestions. We have included a time-frequency depiction under Essential Revision #4 (please refer to our response there for more details). We have added these analyses also as Figure 2—figure supplement 3 in our revised manuscript. In addition, we analyzed the effect of N20 amplitudes on perceived stimulus intensity also within α quintiles (“N20 amplitudes stratified for α power”). Since Signal Detection Theory requires further partitioning within pre-stimulus α quintiles, only few trials would remain for the calculation of the individual *sensitivity d’* and *criterion c* values (e.g., around 66 trials when stratifying for α amplitudes with 5 bins and comparing the first and last tertile (i.e., three bins) of N20 amplitudes within each α quintile, or 40 trials when using quintiles for N20 amplitudes as well). This means that on average across participants only 9.38 trials of the category “weak stimulus perceived as strong stimulus” could be included in such stratified SDT analysis. Since this trial number appears rather small to obtain robust statistical results, we instead used linear-mixed-effects models in close correspondence to the sub-models used in our SEM analysis: Within every pre-stimulus α quintile, we regressed perceived stimulus intensity on N20 amplitude while including the presented stimulus intensity as covariate and modelling random intercepts for every participant. Across α quintiles, the effect of N20 amplitude on the perceived intensity showed a consistent direction, however significance was reached only in the fourth quintile, while the second and fifth quintile showed trends, and the first and third quintile did not show any effect, *z_bin1_*=1.045, *p_bin1_*=.296, *z_bin2_*=1.744, *p_bin2_*=.081, *z_bin3_*=0.476, *p_bin3_*=.634, *z_bin4_*=2.01, *p_bin4_*=.045, *z_bin5_*=1.792, *p_bin5_*=.073, from first to fifth quintile, respectively. Taken together, these results suggest that a similar relation between N20 amplitudes and behavioral responses can be qualitatively observed across different levels of pre-stimulus α amplitude. However, statistical power seems to be lacking to detect these rather small effects when only analyzing subsets of the data (in this case quintiles). In addition, we would like to mention that our SEM approach actually already reflected a similar rationale: Although we did not stratify for different α amplitude levels there, we did include α amplitude as a co-predictor in the analysis of the effect of N20 amplitude on perceived stimulus intensity (see Table 2, fourth equation). Therefore, the N20 effects in the SEM reflected additional effects beyond the effects of pre-stimulus α amplitude, or, in other words, the N20 effects remained when partialing out effects of pre-stimulus α amplitude.

– Furthermore, it would improve readability if the authors used a streamlined statistical approach also for the "thalamic" and "late" evoked responses.

In line with Essential Revision #3, we have additionally run binning analyses for the “later” SEP (i.e., N140 component) and the thalamic SEP (i.e., P15 component), to make a direct comparison also with Signal Detection Theory (SDT) parameters possible: For the N140 component, there was a significant effect on *criterion c*, *t*(31)=-3.010, *p*=.005, but no effect on *sensitivity d’*, *t*(31)=0.246, *p*=.807. For the P15 component, no effects emerged either for *criterion c* or *sensitivity d’*, *t*(12)=1.201, *p*=.253, and *t*(12)=-0.201, *p*=.844, respectively. These findings correspond well to the previous LME analyses and may indeed further facilitate the comparison with the findings for the N20 potential and pre-stimulus α activity. Therefore, we have added these complimentary analyses to our manuscript in the following way:

Results:

“In addition, the SDT analysis based on binning of the P15 amplitudes into quintiles neither suggested a relation with *criterion c* nor with *sensitivity d’*, *t*(12)=1.201, *p*=.253, and *t*(12)=-0.201, *p*=.844, respectively.” (page 14, lines 241 ff.)

and

“These findings were in line with a separate SDT analysis: N140 amplitudes were associated with an effect on *criterion c*, *t*(31)=-3.010, *p*=.005, but no effect on *sensitivity d’* emerged, *t*(31)=0.246, *p*=.807.” (page 15, lines 263 ff.)

Discussion:

“Crucially, our data are at the same time consistent with previous studies on somatosensory processing at later stages, where larger EEG potentials are typically associated with a stronger percept of a given stimulus (e.g., Al et al., 2020; Schröder et al., 2021; Schubert et al., 2006), as both our SDT and LME analyses of the N140 component showed.” (page 19, lines 367 ff.)

and

“Yet, neither our SDT analyses nor the LME models of the thalamus-related P15 component supported this notion.” (page 21, lines 414 ff.)

Methods (page 32, lines 681 ff.):

“The effects of the EEG measures *pre-stimulus α amplitude*, *N20 peak amplitude*, *P15 mean amplitude*, and *N140 mean amplitude* on the SDT measures *sensitivity d’* and *criterion c* were examined using a binning approach: […]”

– The authors convincingly show that evoked peripheral responses are not linked to N20 amplitude. It would be interesting to know whether variability in ongoing peripheral activity is linked to ongoing α activity (i.e. in prestimulus period).

Indeed, this is an interesting aspect and an important sanity check of our dataset. We examined this by two random-slope linear-mixed-effects models with pre-stimulus α amplitude as predictor of CNAP and CMAP amplitude, respectively, while including presented stimulus intensity in both models as covariate. Neither for CNAP nor CMAP, we found a significant effect, *t*(30.296)=-1.334, *p*=.192, and *t*(29.918)=-0.377, *p*=.709, respectively. Additionally, we compared two more alternative SEMs – including the effects *CMAP ~ prestim*, and *CNAP ~ prestim*, respectively – to our original SEM (“SEM 1”). Also here, the model comparisons suggested that the original model without the relations between pre-stimulus α activity and CMAP or CNAP was to be preferred, as indicated by non-significant χ^2^-difference tests (but higher model complexities), *χ^2^diff*=.220, *p*=.639, and *χ^2^diff*=2.342, *p*=.126, respectively. Therefore, ongoing oscillatory brain activity in the α range was not related to our measures of peripheral nerve activity. We have added the SEM comparisons and their interpretation to our revised manuscript in Table 1 (page 12), as well as in the text accordingly:

“To evaluate the model fit, we compared a list of alternative models in- or excluding relevant effect paths (Table 1). As indicated by Chi-Square Difference Tests, the log-likelihood of SEM 1 did not differ from those of SEMs 2-4, 9, and 10. Seeking model parsimony, SEM 1 is preferred over SEMs 2-4, 9, and 10 since the latter alternative models included one more parameter each, while fitting the data equally well.” (page 11, lines 174 ff.)

and:

“Also, peripheral activity in CMAP and CNAP did not show any association with pre-stimulus α activity.” (page 11, lines 197 ff.)

– At the moment the value of the physiological model developed in the discussion is not really clear (see previous comments). It also seems to be particularly geared to the N20 results and leaves open the question whether it also encompasses the "thalamus" and "late" results.

Thank you for pointing out that this aspect was not clear: We aimed to address with our physiological model only the initial cortical portion of the SEP (i.e., the N20 response) because only this SEP component is thought to correspond to “pure” excitation (excitatory post-synaptic potentials). For later SEP components, there will almost always be a mixture of both excitatory and inhibitory contributions whose mutual effects on the SEP amplitude are difficult to disentangle then. Yet, the proposed mechanism of reduced or increased trans-membrane currents leading to smaller or larger EEG amplitudes may be applicable also to other scenarios. Nevertheless, we would like to be careful with such further assumptions as our data is only consistent with the proposed model for N20 responses. We have tried to clarify these considerations in the following way in the Discussion (page 19, lines 373 ff.):

“In this context, it should be further emphasized that our proposed physiological model may be directly applicable to well isolated neural signals only (such as the N20 component of the SEP). The physiological interpretation of amplitudes of later EEG potentials, such as the N140, however, is not as straightforward as described above, since several distinct SEP components may interact (Auksztulewicz et al., 2012), and excitatory and inhibitory contributions cannot be readily distinguished.”

References

Berens, P. (2009). CircStat A MATLAB Toolbox for Circular Statistics. Journal of Statistical Software, 31.

Haegens, S., Barczak, A., Musacchia, G., Lipton, M. L., Mehta, A. D., Lakatos, P., and Schroeder, C. E. (2015). Laminar Profile and Physiology of the α Rhythm in Primary Visual, Auditory, and Somatosensory Regions of Neocortex. The Journal of Neuroscience, 35, 14341–14352.

Hanslmayr, S., Klimesch, W., Sauseng, P., Gruber, W., Doppelmayr, M., Freunberger, R., Pecherstorfer, T., and Birbaumer, N. (2007). Α phase reset contributes to the generation of ERPs. Cerebral cortex (New York, N.Y. 1991), 17, 1–8.

Iemi, L., Chaumon, M., Crouzet, S. M., and Busch, N. A. (2017). Spontaneous Neural Oscillations Bias Perception by Modulating Baseline Excitability. The Journal of Neuroscience, 37, 807–819.

Morey, R. D. (2008). Confidence Intervals from Normalized Data: A correction to Cousineau (2005). Tutorial in Quantitative Methods for Psychology, 4, 61–64.

Samaha, J., Iemi, L., Haegens, S., and Busch, N. A. (2020). Spontaneous Brain Oscillations and Perceptual Decision-Making. Trends in cognitive sciences, 24, 639–653.

Sauseng, P., Klimesch, W., Gruber, W. R., Hanslmayr, S., Freunberger, R., and Doppelmayr, M. (2007). Are event-related potential components generated by phase resetting of brain oscillations? A critical discussion. Neuroscience, 146, 1435–1444.